# Physical versus economic water footprints in crop production: a spatial and temporal analysis for China

Xi Yang[1, 3], La Zhuo[2, 3, 4], Pengxuan Xie[1, 3], Hongrong Huang[1, 3], Bianbian Feng[1, 3], Pute Wu[2, 3, 4]

[1]College of Water Resources and Architectural Engineering, Northwest A&F University, Yangling 712100, China

[2]Institute of Soil and Water Conservation, Northwest A&F University, Yangling 712100, China

[3]Institute of Water-saving Agriculture in Arid Regions of China, Northwest A&F University, Yangling 712100, China

[4]Institute of Soil and Water Conservation, Chinese Academy of Sciences and Ministry of Water Resources, Yangling 712100, China

*Correspondence to*: La Zhuo (zhuola@nwafu.edu.cn); Pute Wu (gjzwpt@vip.sina.com)

## Abstract

A core goal of sustainable agricultural water resources management is to implement lower water footprint (WF), i.e., higher water productivity, while maximising economic benefits in crop production. However, previous studies mostly focused on crop water productivity from a single physical perspective. Little attention is paid to synergies and trade-offs between water consumption and economic value creation of crop production. Distinguishing between blue and green water composition, grain and cash crops, and irrigation and rainfed production modes in China, this study calculates the production-based WF (PWF) and derives the economic value-based WF (EWF) of 14 major crops in 31 provinces for each year over 2001-2016. The synergy evaluation index (SI) of PWF and EWF is proposed to reveal the synergies and trade-offs of crop water productivity and its economic value from the WF perspective. Results show that both the PWF and EWF of most considered crops in China decreased with the increase of crop yield and prices. The high (low) values of both PWF and EWF of grain crop tended to obvious cluster in space and there existed a huge difference between blue and green water in economic value creation. Moreover, the SI revealed a serious incongruity between PWFs and EWFs both in grain and cash crops. Negative SI values occurred mostly in northwest China for grain crops, and overall more often and with lower values for cash crops. Unreasonable regional planting structure and crop prices resulted in this incongruity, suggesting the need to promote regional coordinated development to adjust the planting structure according to local conditions and to regulate crop prices rationally.

## 1 Introduction

Humanity is facing the increasingly severe threat of water shortage and accompanying rising food risks (Mekonnen and Hoekstra, 2016; Veldkamp et al., 2017), posing great challenges to agricultural water resource management. The economic benefits of water use form one important pillar of fresh water distribution (Hoekstra, 2014). However, traditional studies on agricultural efficient water use focus on crop water productivity from the physical perspective, and rarely make comprehensive

evaluations combining the results with an economic perspective. The water footprint (WF) (Hoekstra, 2003) reveals the consumption and pollution of water in the process of production or consumption and assesses fresh water appropriation in its entirety (Hoekstra et al., 2011). The consumptive WF of crop production can be divided into blue and green WFs (Hoekstra et al., 2011). Blue water is surface and ground water, whereas green water is defined as the water kept in the unsaturated soil layer and precipitation, which is eventually transferred into canopy evapotranspiration (Falkenmark and Rockstrom, 2006). In agriculture, the blue WF measures irrigation water consumption. Green WF refers to the consumption of rainwater (Hoekstra et al., 2011). As comprehensive index to evaluate types, quantities, and efficiency of water use in the process of crop production, the WF of crop production can be expressed based on both production (PWF, $m^3$ $kg^{-1}$) and economic value (EWF, $m^3$ per monetary unit) (Garrido et al., 2010; Hoekstra et al., 2011), which unifies the measurement of the physical and economic levels. PWF and EWF provide clear insights for reducing the water resources input for harvesting crop yields and optimizing the economic benefits per unit of water consumption, respectively.

Garrido et al. (2010) firstly evaluated WF in terms of $m^3$ $€^{-1}$, from a perspective of hydrology and economy for agricultural production of Spain. They found that in areas where blue water was scarce but dominant in crop production, the scarce blue water resource was used to irrigate high-value crops, thus achieving higher yields and economic benefits, with a more efficient blue water utilisation with increasing scarcity. In a case study for Kenya, Mekonnen and Hoekstra (2014a) encouraged to use domestic water resource for production of the rainfed cash crops with high economic benefits, rather than for water-intensive export commodities with low economic benefits. Schyns and Hoekstra (2014) found that water and land resources in Morocco were mainly used to produce export crops with relatively low economic value (in terms of USD $m^{-3}$ and USD $ha^{-1}$), and that water-scarce countries should attribute great importance to the allocation of freshwater and adjust crop planting structure from the perspective of economic efficiency. Chouchane et al. (2015) quantified the WF in Tunisia and evaluated the blue and green economic water productivity and economic land productivity in irrigation and rainfed agriculture from an economic perspective. They showed that irrigation water was not generally used to increase economic water productivity (USD $m^{-3}$) but rather to increase economic land productivity (USD $ha^{-1}$), so it would be advantageous to expand the irrigated area of crops with high economic water productivity. Furthermore, in recent years, there have been studies on the dairy industry (Owusu-Sekyere et al., 2017a; Owusu-Sekyere et al., 2017b),the meat industry (Ibidhi and Salem, 2018) and the wine industry (Miglietta et al., 2018) to explore the WF assessment combined with an economic perspective.

Nevertheless, the above studies lacked a complete temporal and spatial evolution analysis of the WF from the economic perspective. More importantly, the above studies did not involve the study of WF coordination in different aspects, which means a good synergy in reducing the water resources input for harvesting crop yields and optimizing the economic benefits per unit of water consumption, compared with national average level. Thus, the synergies and trade-offs between water consumption and economic value creation during crop production in WF assessment, which is undoubtedly of great significance, are ignored.

Scientifically planning agricultural water resource utilisation and balancing crop production, water consumption and social
economic development are severe challenges faced by all humankind. However, China, with millions of small farmers led by
smallholder production, has become one of the regions facing the biggest challenges (Tilman et al., 2011; Gao and Bryan,
2017; Cui et al., 2018). Being the country with the largest population and food consumption, China faces a series of problems,
such as extensive management and low utilisation rate of water resource in agricultural production (Khan et al., 2009; Kang
et al., 2017). Previous studies on China have quantified the WF of crop production at the irrigation district scale (Sun et al.,
2013; Cao et al., 2014; Sun et al., 2017), watershed scale (Zhuo et al., 2014; Zhuo et al., 2016c) and national scale (Zhuo et
al., 2016b; Wang et al., 2019). Sun et al. (2013) found that the WF of crop depended on agricultural management rather than
on regional climate differences; Zhuo et al. (2016b) showed that China's domestic food trade was determined by the economy
and government policies, not by regional differences in water endowments; Wang et al. (2019) showed possibility and
importance of accounting for developments of water-saving techniques in largescale crop WF estimations. However, most of
these studies focused on quantifying WF from a single physical perspective. To our knowledge, there is no study yet to provide
clear insights into the economic benefits of water use.
To fill the above research gap, the current study objective is, taking China over 2001-2016 as the study case, to explore the
relationship between water resource consumption and economic value creation of intra-national scale crop production, and to
propose a synergy evaluation index (SI) of PWF and EWF to reveal the synergies and trade-offs of crop water productivity
and its economic value from the WF perspective. First, the blue and green PWF ($PWF_b$, $PWF_g$) of 14 major crops (winter
wheat, spring wheat, spring maize, summer maize, rice, soybean, cotton, groundnut, rapeseed, sugar beet, sugarcane, citrus,
apple, and tobacco) is calculated annually in 31 provinces at the meteorological station level, and the corresponding EWF is
derived. Second, crops are distinguished between grain and cash crops, with Mann-Kendall trend test and spatial
autocorrelation analysis method for evaluation of the temporal and spatial evolution characteristics of PWF and EWF. Finally,
the synergy evaluation index (SI) is constructed. Consequently, based on the quantification of PWF and EWF, we constructed
the synergy evaluation index (SI) of water footprint, so that the original intention of the study which is a comprehensive
assessment from the perspective of both physics and economics can be implemented.
**2 Method and data**
**2.1 AquaCrop modeling**
Crop WF per unit mass is defined by the evapotranspiration (ET) and yield (Y) over the growing period (Hoekstra et al., 2011).
The AquaCrop model (Hsiao et al., 2009; Raes et al., 2009; Steduto et al., 2009), a water driven crop water productivity model
developed by FAO, is used to simulate the daily green and blue ET and yield Y of 14 crops for each station. The AquaCrop
has fewer parameters than other crop growth models and provides a better balance between simplicity, accuracy, and robustness
(Steduto et al., 2009). A large number of studies have demonstrated the good performance of AquaCrop in simulating crop
growth and water use under different environmental conditions (Abedinpour et al., 2012; Jin et al., 2014; Kumar et al., 2014).
Also, there have been a number of studies using AquaCrop to calculate water footprints (Chukalla et al., 2015; Zhuo et al.,
2016a; Zhuo et al., 2016c; Wang et al., 2019). The dynamic soil water balance in the AquaCrop model is shown in Eq. (1):
$$S_{[t]} = S_{[t-1]} + PR_{[t]} + IRR_{[t]} + CR_{[t]} - ET_{[t]} - RO_{[t]} - DP_{[t]}, \tag{1}$$
where $S_{[t]}$ (mm) is the soil moisture content at the end of day t; $PR_{[t]}$ (mm) is the rainfall on day t; $IRR_{[t]}$ (mm) is the irrigation
amount on day t; $CR_{[t]}$ (mm) is the capillary rise from groundwater; $RO_{[t]}$ (mm) is the surface runoff generated by rainfall and
irrigation on day t; $DP_{[t]}$ (mm) is the amount of deep percolation on day t. $RO_{[t]}$ is obtained through the Soil Conservation
Service curve-number equation (USDA, 1964; Rallison, 1980; Steenhuis et al., 1995):
$$RO_{[t]} = \frac{(PR_{[t]} - I_a)^2}{PR_{[t]} + S - I_a}, \tag{2}$$
where $S$ (mm) is the maximum potential storage, which is a function of the soil curve number; $I_a$ (mm) is the initial water loss
before surface runoff; $DP_{[t]}$ (mm) is determined by the drainage capacity (m$^3$ m$^{-3}$ day$^{-1}$). When the soil water content is less
than or equal to the field capacity, the drainage capacity is zero (Raes et al., 2017).
AquaCrop model is able to track the daily inflow and outflow at the root zone boundary. On this basis, we use the blue and
green WF calculation framework by Chukalla et al. (2015) , Zhuo et al. (2016c) and Hoekstra (2019) combined with the model
of soil water dynamic balance to separate the daily blue and green ET (mm), as shown in Eqs. (3) and (4):
$$S_{b[t]} = S_{b[t-1]} + IRR_{[t]} - RO_{[t]} \times \frac{IRR_{[t]}}{PR_{[t]} + IRR_{[t]}} - (DP_{[t]} + ET_{[t]}) \times \frac{S_{b[t-1]}}{S_{[t-1]}}, \tag{3}$$
$$S_{g[t]} = S_{g[t-1]} + PR_{[t]} - RO_{[t]} \times \frac{PR_{[t]}}{PR_{[t]} + IRR_{[t]}} - (DP_{[t]} + ET_{[t]}) \times \frac{S_{g[t-1]}}{S_{[t-1]}}, \tag{4}$$
where $S_{b[t]}$ and $S_{g[t]}$ (mm) respectively represent the blue and green soil water content at the end of day t. According to Siebert
and Döll (2010), the maximum soil moisture of rainfed fallow land two years before planting is taken as the initial soil moisture
for simulating. At the same time, the initial soil water during the growing period is set as green water (Zhuo et al., 2016c).
The blue and green components in DP and ET were calculated per day based on the fractions of blue and green water in the
total soil water content at the end of the previous day (Zhuo et al., 2016a), which are shown in Eqs. (5) and (6):
$\quad ET_{b[t]} = ET_{[t]} \times \dfrac{S_{b[t-1]}}{S_{[t-1]}}$ , $\hspace{11cm}$ (5)
$\quad ET_{g[t]} = ET_{[t]} \times \dfrac{S_{g[t-1]}}{S_{[t-1]}}$ , $\hspace{11cm}$ (6)
Using the normalized biomass water productivity (WP*, kg m$^{-2}$), which is normalized for the atmospheric carbon dioxide ($CO_2$)
concentration, the evaporative demand of the atmosphere ($ET_0$) and crop classes (C3 or C4 crops), AquaCrop calculates daily
aboveground biomass production (B, kg) from daily transpiration (Tr) and the corresponding daily reference
evapotranspiration ($ET_0$) (Steduto et al., 2009):
$\quad B = WP * \sum \dfrac{T_{r[t]}}{ET_{0[t]}}$ $\hspace{11cm}$ (7)
The crop yield (harvested biomass) is the product of the above-ground biomass (B) and the adjusted reference harvest index
($HI_0$, %) (Raes et al., 2017).
$\quad Y = f_{HI} HI_0 B$ , $\hspace{11cm}$ (8)
where the adjustment factor($f_{HI}$) reflects the water and temperature stress depending on the timing and extent during the crop
cycle.

**2.2 Calculation of production-based water footprint (PWF)**

The PWF of 14 major crops were calculated annually in 31 provinces at the meteorological station level. Table 1 shows the
number of meteorological stations per province. The PWF (m$^3$ kg$^{-1}$) consists of the blue PWF (PWF$_b$, m$^3$ kg$^{-1}$) and the green
PWF (PWF$_g$, m$^3$ kg$^{-1}$), which are respectively calculated from the daily blue evapotranspiration ($ET_{b[t]}$, mm) and daily green
evapotranspiration ($ET_{g[t]}$, mm) and crop yield (Y, kg ha$^{-1}$) during the growing period (Hoekstra et al., 2011), as shown in Eqs.
132 (9) - (11):

$\quad PWF = PWF_b + PWF_g$ , $\hspace{10cm}$ (9)
$\quad PWF_b = \dfrac{10 \times \sum\limits_{t=1}^{gp} ET_{b[t]}}{Y}$ , $\hspace{10cm}$ (10)
$$PWF_g = \frac{10 \times \sum\limits_{t=1}^{gp} ET_{g[t]}}{Y} , \qquad (11)$$
where gp (day) is the length of growing period; 10 is the conversion coefficient. The daily ET and Y values during the growth
period are simulated by the AquaCrop model. Being consistent with the existing calibration method which has been widely
applied (Mekonnen and Hoekstra, 2011; Zhuo et al., 2016b; Zhuo et al., 2016c; Wang et al., 2019; Zhuo et al., 2019), the
modeled crop yield was calibrated at provincial level according to the statistics (NBSC, 2019). Within a province, we calibrated
the average level of the modeled yields among station points to match the provincial statistics. Therefore, we kept the spatial
variation in crop yields, so that in associated water footprints simulated by AquaCrop model.

Table 1. Number of meteorological stations per province.

**2.3 Calculation of economic value-based water footprint (EWF)**
Following Hoekstra et al. (2011), the EWF ($m^3$ $USD^{-1}$) of crop production represents the water consumption per unit of
economic value.
$$EWF = \frac{PWF}{UP} , \qquad (12)$$
where PWF ($m^3$ $kg^{-1}$) the production-based WF, and UP (USD $kg^{-1}$) the crop unit price. The economic benefit unit refers to
crop price in the current study. The EWF is numerically equal to the inverse of the economic water productivity. Considering
the PWF and the EWF together provides a clear and intuitive measurement to analyse the synergy relationship between water
consumption of crop production and economic value creation. To eliminate the influence of inflation, we use the consumer
price index (CPI) to calculate the inflation rate of China based on 2001 and to convert the annual crop current price into the
2001 constant Chinese Yuan price (Constant 2001 CNY). Then, we convert it to the 2001 constant American dollar price
(Constant 2001 USD).
Referring to Chouchane et al. (2015), when calculating the blue and green EWF, we distinguish between irrigation and rainfed
agricultural modes. In rainfed agriculture, the green EWF ($EWF_{g,rf}$) is obtained by dividing the green water consumption per
unit yield under rainfed condition by the unit price of crops, as shown in Eq. (13). Compared to rainfed agriculture, the ratio
of crop yield increment under full irrigation is obtained by AquaCrop model. We use it to distinguish the blue and green EWF
in irrigation agriculture ($EWF_{b,ir}$, $EWF_{g,ir}$), as shown in Eqs. (14) - (16):
$$EWF_{g,rf} = \frac{CWU_{g,rf}}{Y_{RF} \times UP},$$ (13)
$$\alpha = \frac{Y_{IR} - Y_{RF}}{Y_{IR}},$$ (14)
$$EWF_{b,ir} = \frac{CWU_{b,ir}}{Y_{IR} \times UP \times \alpha},$$ (15)
$$EWF_{g,ir} = \frac{CWU_{g,ir}}{Y_{IR} \times UP \times (1-\alpha)},$$ (16)
where $CWU_{g,rf}$ (m$^3$ ha$^{-1}$) represents the consumption of green water per unit area in rainfed agriculture; $CWU_{b,ir}$ (m$^3$ ha$^{-1}$) and
$CWU_{g,ir}$ (m$^3$ ha$^{-1}$) represent the consumption per unit area in irrigation agriculture of blue and green water, respectively; $\alpha$ is
the ratio of crop yield increment under full irrigation obtained by AquaCrop model; $Y_{RF}$(kg ha$^{-1}$) and $Y_{IR}$(kg ha$^{-1}$) represent
the simulated crop yield after calibrated at provincial level under the rainfed and irrigation modes, respectively. The $EWF_{g,rf}$
represents the amount of green water consumption per economic benefit unit in rainfed agriculture (also refers to the amount
of green water input for each additional economic benefit unit); $EWF_{b,ir}$ ($EWF_{g,ir}$) refers to the additional amount of blue (green)
water for each additional unit economic benefit under the same green (blue) water input in irrigation agriculture.

## 172 2.4 Spatial and temporal evolution of WFs

The Mann-Kendall (M-K) trend test (Mann, 1945; Kendall, 1975) is used to test the annual variation trend of WF of crop
production from 2001 to 2016. When using M-K test for trend analysis, the null hypothesis $H_0$ is the that all variables in WF
time series $\{WF_i \mid i = 1, 2, ..., 16\}$ are independent and identical in distribution, with no variation trend; the alternative hypothesis
$H_1$ is that all $i$, $j \le 16$ and $i \ne j$, in the distribution of $WF_i$ and $WF_j$ are different, with an obvious upward or downward trend in
the sequence. The M-K statistic S is shown in Eq. (17):
$$S = \sum_{i=1}^{n-1} \sum_{j=i+1}^{n} sgn(WF_j - WF_i),$$ (17)
where WF$_j$ and WF$_i$ are the data values of year j and i of the WF time series, respectively; n is the length of the data sample,
16; sgn is sign function, depicted in Eq. (18).
$$sgn(\theta) = \begin{cases} 1 & \theta > 0 \\ 0 & \theta = 0 \\ -1 & \theta < 0 \end{cases}.$$ (18)
When n ≥ 8, the M-K statistic S roughly follows a normal distribution, whose mean value is zero, and the variance can be
calculated by Eq. (19).
$$Var(S) = \frac{n(n-1)(2n+5) - \sum_{p=1}^{g} t_p(t_p - 1)(2t_p + 5)}{18},$$ (19)
where g is the number of tied groups, and $t_p$ is the number of data values in the $P^{th}$ group (Kisi and Ay, 2014). When n >10,
the test statistic $Z_c$ converges to the standard normal distribution, which is calculated by Eq. (20).
$$Z_c = \begin{cases} (S-1)/\sqrt{Var(S)} & S > 0 \\ 0 & S = 0 \\ (S+1)/\sqrt{Var(S)} & S < 0 \end{cases}.$$ (20)
Using two-tailed test, when the absolute value of $Z_c$ exceeds 1.96 and 2.58, it means that the significance test of 95% and 99%
has been passed, respectively. The positive $Z_c$ indicates an upward trend, while a negative value means a downward trend.
The first law of geography states that everything is related, and things close to each other are more relevant (Tobler, 1970).
The global and local spatial relevance of WF is expressed by the index Moran's I (Moran, 1950). A positive spatial
autocorrelation exists, when the high or low values of the feature variables of adjacent regions show a clustering tendency in
space; and a negative spatial autocorrelation means that the value of the feature variables of adjacent regions is opposite to that
of the variable of the examined region. The Global Moran's I is used to evaluate the overall spatial relevance of WF of crop
production, shown in Eq. (21).
$$I = \frac{n \sum_{i=1}^{n} \sum_{j=1}^{n} W_{ij}(WF_i - \overline{WF})(WF_j - \overline{WF})}{\sum_{i=1}^{n}(WF_i - \overline{WF})^2 \sum_{i=1}^{n} \sum_{j=1}^{n} W_{ij}},$$ (21)
where n is the number of provinces, 31; $WF_i$ is crop WF of province i; $\overline{WF}$ is the average WF; and $W_{ij}$ is the spatial weight
between the province i and j, which represents the potential interaction forces between the spatial units. When province i and
j are adjacent, $W_{ij}=1$; when not adjacent, $W_{ij}=0$. At the given significance level (0.05 in this study), if the Global Moran's I is

significantly positive, it indicates that provinces with similar geographical attributes are clustered in space. On the contrary, if the Global Moran's I is significantly negative, it means that provinces with different geographical attributes are clustered in space. Local Moran index (LISA) (Anselin, 1995) is used to detect whether there is local clustering of attributes, and the level (high or low) of the WF of a province is shown by the LISA cluster map. The LISA cluster map contains four types (Anselin, 2005): high-high (H-H) and low-low (L-L) indicate that the level (high or low) of WF in this province is consistent with adjacent provinces; high-low (H-L) and low-high (L-H) mean that the level (high or low) of WF in this province is opposite to adjacent provinces. The analysis of spatial autocorrelation can be realised by the GeoDa. The GeoDa is a free software program intended to serve as a user-friendly and graphical introduction to spatial analysis. It includes functionality ranging from simple mapping to exploratory data analysis, the visualization of global and local spatial autocorrelation, and spatial regression. A key feature of the GeoDa is an interactive environment that combines maps with statistical graphics, using the technology of dynamically linked windows. In terms of the range of spatial statistical techniques included, the GeoDa is most alike to the collection of functions developed in the open-source R environment (Anselin et al., 2006).

**2.5 The synergy evaluation index (SI) of PWF and EWF**

The synergy evaluation index (SI) in the current study is the measure of the synergy levels between the PWF and EWF of crops, by summing up their corresponding difference between the water footprint and the base value divided by the range (the maximum minus the minimum) of the water footprint. Here, we adopt the national average level water footprint value as the reference for comparison. The SI is calculated as follows:

$$SI_{i,j,c} = \frac{\overline{PWF_{j,c}} - PWF_{i,j,c}}{PWF_{j,c,max} - PWF_{j,c,min}} + \frac{\overline{EWF_{j,c}} - EWF_{i,j,c}}{EWF_{j,c,max} - EWF_{j,c,min}} , \tag{22}$$

where $SI_{i,j,c}$ is the synergy evaluation index of PWF and EWF of crop c at province i in year j, $\overline{PWF_{j,c}}$ (m$^3$ kg$^{-1}$) and $\overline{EWF_{j,c}}$ (m$^3$ USD$^{-1}$) are the averages at the national level in year j. Obviously, the absolute value of the difference between the WF and their corresponding national average level cannot exceed the maximum minus minimum values. Therefore, the absolute value of SI cannot exceed 2. When the PWF and EWF in a region are both lower than the respective average at the national level, the SI of the region must be positive; when the PWF and EWF in a region are both higher than the respective average at the national level, the SI of the region must be negative. When one is higher, and the other is lower than the corresponding average, the SI may be positive or negative, depending on the difference between the provincial value and the national average.

**2.6 Data sources**

The planting area and yield data of each province were obtained from NBSC (2019). The provincial price data of crops were obtained from the China National Knowledge Infrastructure (CNKI, 2019). The current crop prices were converted to the constant prices using the inflation rate based on 2001. The consumer price index (CPI), which is used to calculate the inflation

rate, was retrieved from NBSC (2019). The exchange rate used to convert local constant prices into American constant prices was taken from The World Bank (2019). The meteorological data on daily precipitation, daily mean maximum temperature and daily mean minimum temperature required for the AquaCrop model of 698 meteorological stations in the study area (see Fig.1) were downloaded from CMDC (2019). The irrigation and rainfed areas of crops were retrieved from MIRCA2000 (Portmann et al., 2010). The soil texture data were taken from the ISRIC database (Dijkshoorn et al., 2008). The soil water content data were from Batjes (2012); The dates of planting of crops referred to (Chen et al., 1995). The harvest indexes were taken from Xie et al. (2011) and Zhang and Zhu (1990). Crop growth periods and maximum root depths were taken from Allen et al. (1998) and Hoekstra and Chapagain (2007).

Figure 1. Considered weather stations across mainland China.

## 3 Results

### 3.1 Temporal and spatial evolution of PWF

At the national average level, the PWF of both grain and cash crops showed a significant downward trend over the study period 2001-2016. With the increase of crop yield (grain crop increasing by 26%, cash crop increasing by 62%), the PWF of grain crop decreased by 20% from 1.16 $m^3$ $kg^{-1}$ to 0.93 $m^3$ $kg^{-1}$ (Fig. 2a); and the PWF of cash crop decreased by 35% from 0.70 $m^3$ $kg^{-1}$ to 0.46 $m^3$ $kg^{-1}$ (Fig. 2b). As for the composition of the WF, the proportion of blue WF of crop production showed a decreasing trend. The proportion of blue WF of grain and cash crops decreased from 39% and 17% in 2001 to 34% and 14% in 2016, respectively.

Figure 2. Interannual variability of national average production-based water footprint (PWF) of (a) grain and (b) cash crops in China over 2001-2016.

Table 2 lists the PWF, yield and blue and green water consumption by crops under irrigated and rainfed agriculture in 2001 and 2016. Concerning grain crops, soybean had the highest PWF (2.79 $m^3$ $kg^{-1}$ in 2016), followed by spring wheat (1.51 $m^3$ $kg^{-1}$ in 2016). Rice had the lowest PWF (0.78 $m^3$ $kg^{-1}$ in 2016). Among cash crops, cotton had the highest PWF (3.68 $m^3$ $kg^{-1}$ in 2016), while sugar beet consumed the least water per yield (0.06 $m^3$ $kg^{-1}$ in 2016). The proportion of blue WF in spring

wheat was the highest (69% in 2016). Cotton had the highest proportion of blue WF (32% in 2016) in cash crops. Winter wheat
is the grain crop with the highest output in China, and its PWF decreased by 29% (from 1.47 $m^3$ $kg^{-1}$ in 2001 to 1.04 $m^3$ $kg^{-1}$
in 2016). Cotton is the cash crop with the highest water consumption per yield, and its PWF decreased by 31% (from 5.29 $m^3$
$kg^{-1}$ in 2001 to 3.68 $m^3$ $kg^{-1}$ in 2016). The M-K test results of each crop's PWF in Table 3 further confirm the above views. The
PWF and yield of different crops had different temporal evolutions. The temporal trends in the $PWF_b$ and $PWF_g$ of a same
crop were also different. Among grain crops, winter wheat had the lowest M-K statistical value in PWF (-4.547) and the highest
in yield (5.178) jointly showing an obvious positive trend on improving water use efficiency. While the M-K statistic value of
soybean was only -0.675, which meant that the PWF of soybean had little decrease. Soybean planting was dominated by
individual farmer mode, with small and fragmented scales and a low planting mechanization degree. Moreover, the harvested
area was shrunk (7,202 thousand hectares in 2016, 24% less than 2001). For cash crops, the changes of PWF and yield were
most pronounced for fruit crops (apple and citrus). The M-K test result of $PWF_b$ of cotton with highest water consumption
intensity was zero, with almost no changes, given little changes in the yield level at most cotton growing areas.

Table 2. National average production-based water footprint (PWF) and economic value-based water footprint (EWF) of crops
in China for the years 2001 and 2016.

Table 3. M-K analysis of production-based water footprint (PWF) and economic value-based water footprint (EWF) of crops
in China.

Figure 3a and 3b show the spatial distribution of PWF of grain and cash crops across 31 provinces, respectively, in four
representative years (2001, 2006, 2011 and 2016). The PWF of grain crop was overall higher in northwest of China, represented
by provinces Shaanxi, Gansu and Ningxia, with the phenomenon of clustered distribution. The south-eastern coastal areas such
as Guangdong, Fujian and Zhejiang were at a relatively low level. The main reason behind is that the drier northwest, where
grows wheat and maize, has relatively higher evapotranspiration so that higher PWF. While the water-abundant and wet
southeast coastal provinces grow rice with a lower PWF. Consistently with the national level analysis, the PWF of the 31
provinces decreased significantly over time (Fig. 3a). Specifically, in north-western China, Gansu province, where the water-
intensive wheat and maize were the main grain crops (wheat and maize accounting for 95% of grain crops in 2016), had the
largest grain crop PWF (mean 1.43 $m^3$ $kg^{-1}$) and showed an obvious downward trend, which decreased by 30% from 1.73 $m^3$
$kg^{-1}$ in 2001 to 1.21 $m^3$ $kg^{-1}$ in 2016. Concerning the composition of blue and green water, Xinjiang had the largest proportion

of blue water in grain crops among the 31 provinces, with annual average of 75%, far higher than the national average (36%); the proportion of blue water in grain production in Jilin province was the smallest, with annual average of 20%.

Differently from grain crop, the PWF of cash crop was higher in the Beijing-Tianjin-Hebei region, and lower in Inner Mongolia province and the southern areas (Guangdong, Guangxi and Hainan), without an obvious clustered characteristic (Fig. 3b). This can be interpreted that regarding the cash crops, the dominant crop differs among provinces which resulted in obvious scattered characteristics in related WFs. For instance, cotton and groundnut with PWF of 3.68 $m^3$ $kg^{-1}$ and 1.49 $m^3$ $kg^{-1}$ (in 2016) were the leading cash crops in Beijing-Tianjin-Hebei region whereas rapeseed of lower PWF (1.04 $m^3$ $kg^{-1}$ in 2016) was the main cash crop in Inner Mongolia. Specifically, during the study period, the PWF of cash crop in Tianjin where cotton was the main cash crop was the largest (3.31 $m^3$ $kg^{-1}$ in 2011, much higher than the national level of 0.51 $m^3$ $kg^{-1}$ the same year), with the annual average of 2.90 $m^3$ $kg^{-1}$. The PWF of cash crop in Guangxi where citrus and sugarcane were dominant was the smallest, with annual average of 0.14 $m^3$ $kg^{-1}$, much lower than the national level of 0.54 $m^3$ $kg^{-1}$. Concerning the composition of blue and green water, the proportion of blue water was larger in northern and north-eastern China, and lower in southern and southwestern China. Among them, the proportion of blue water of cash crop was the largest in Jilin province, with annual average of 35%, while the proportion of blue water in Qinghai province was less than 1%, which was the lowest in China. These results can be explained by the fact that Jilin's main cash crop was groundnut (88% in 2016), with high proportion of blue water consumption, while Qinghai's 99% of cash crops was rainfed rapeseed.

Figure 3. Temporal and spatial evolution of production-based water footprint (PWF) of (a) grain and (b) cash crops in China.

Table 4 shows Global Moran's I of PWF of grain and cash crops. The annual average global Moran's I of PWF of grain crop was 0.263, with a clustered spatial distribution in most provinces, and gradually moderated over time (Moran's I decreased from 0.559 in 2001 to 0.214 in 2016). The spatial pattern of PWF of cash crop did not show obvious agglomeration, and the average Moran's I was only 0.163.

Table 4. Moran's I test for production-based water footprint (PWF) and economic value-based water footprint (EWF) of crop production.

The LISA cluster map shows that the H-H regions of PWF of grain crop gathered in Gansu, Ningxia, Shaanxi and Inner
Mongolia, and the L-L regions gathered in Guangdong, Zhejiang, Fujian, and Jiangxi (Fig. 4a). At the beginning of the study
period, the PWF in 2001 showed an obvious positive spatial correlation, with 13 significant provinces (Gansu, Ningxia,
Shaanxi, Shanxi, Inner Mongolia, and Hebei in H-H regions; Guangdong, Zhejiang, Fujian, Jiangxi, Anhui, Jiangsu, and Hunan
in L-L regions). In time, the H-H regions in north western China gradually decreased, leaving only Ningxia in H-H regions,
while L-L regions remained relatively stable. Overall, there were 7 significant regions in 2016, indicating that the spatial
agglomeration of PWF of grain crop decreased with time. This indicates that with the development of water-saving technology
and the improvement of agricultural water resource management level, the utilization efficiency of agricultural water resources
in the arid northwest region has been gradually improved, while the gap with the more developed and water-rich southern
provinces is narrowing. As for cash crop, no obvious agglomeration existed (Fig. 4b).

Figure 4. The LISA cluster maps of production-based water footprint (PWF) of (a) grain and (b) cash crops.

**3.2 Temporal and spatial evolution of EWF**
Similar to the evolution of PWF, the EWF of both grain and cash crops showed a significant declining trend at the national
average level. With the increase of crop price (grain crop increasing by 40%, cash crop increasing by 70%), the EWF of grain
crop decreased by 44%, from 9.01 $m^3$ $USD^{-1}$ to 5.04 $m^3$ $USD^{-1}$ (Fig. 5a); the EWF of cash crop decreased by 62%, from 5.39
$m^3$ $USD^{-1}$ to 2.05 $m^3$ $USD^{-1}$ (Fig. 5b).
In terms of grain crop, the $EWF_{b,ir}$ fluctuated, reaching the highest value of 25.58 $m^3$ $USD^{-1}$ in 2002 and falling to the lowest
of 12.26 $m^3$ $USD^{-1}$ in 2010. In contrast, the $EWF_{g,ir}$ and $EWF_{g,rf}$ showed a significant and steady declining trend, decreasing
from 5.32 $m^3$ $USD^{-1}$ and 9.05 $m^3$ $USD^{-1}$ in 2001 to 2.96 $m^3$ $USD^{-1}$ and 4.94 $m^3$ $USD^{-1}$ in 2016, respectively. Among the three
types of WF, the $EWF_{g,ir}$ was the lowest (mean 3.11 $m^3$ $USD^{-1}$), $EWF_{b,ir}$ was the highest (mean 15.49 $m^3$ $USD^{-1}$), and $EWF_{g,rf}$
(mean 5.31 $m^3$ $USD^{-1}$) was close to the average EWF (5.41 $m^3$ $USD^{-1}$) in irrigation and rainfed production modes. This suggests
that more water was required per additional benefit unit under irrigation than under rainfed mode, whereas in the irrigated
agriculture, compared with blue water, increasing the input of green water may result in more economic benefits. Therefore,
utilisation efficiency of green water resource for grain crops should be improved.
Concerning cash crop, the $EWF_{b,ir}$ decreased by 50% from 10.54 $m^3$ $USD^{-1}$ to 5.22 $m^3$ $USD^{-1}$. Compared to grain crop, the
difference between the $EWF_{g,ir}$ and $EWF_{g,rf}$ was smaller, with average values of 1.90 $m^3$ $USD^{-1}$ and 2.48 $m^3$ $USD^{-1}$, respectively.
In addition, compared to grain crop, the EWF of cash crop was lower, which indicated that cash crop production could get
more economic benefits per water consumption unit. Besides, increasing the input of green water resource could obtain higher
economic benefits, and the rainfed production had greater economic potential.

Figure 5. Interannual variability of economic value-based water footprint (EWF) of (a) grain and (b) cash crops in China over
345   2001-2016.


Table 2 lists the EWF by crops in 2001 and 2016 at the national scale. Among grain crops, soybean, which consumed the most
water per yield unit (2.79 $m^3$ $kg^{-1}$ in 2016), also had the highest EWF; the second most water-intensive, spring wheat (1.51 $m^3$
$kg^{-1}$ in 2016) had the second highest EWF (8.02 $m^3$ $USD^{-1}$ in 2016); rice, with the lowest water consumption per yield unit
(0.78 $m^3$ $kg^{-1}$ in 2016) also had the lowest EWF (3.39 $m^3$ $USD^{-1}$ in 2016). Regarding cash crops, cotton, with the highest water
consumption per yield unit (3.68 $m^3$ $kg^{-1}$ in 2016) was the crop with the highest EWF (3.00 $m^3$ $USD^{-1}$ in 2016); groundnut's
EWF ranked second (2.76 $m^3$ $USD^{-1}$ in 2016); sugar beet had lowest water consumption per yield unit (0.06 $m^3$ $kg^{-1}$ in 2016),
with an EWF (1.60 $m^3$ $USD^{-1}$ in 2016) much lower than the average EWF of cash crops (2.05 $m^3$ $USD^{-1}$ in 2016).
Sugarcane had the lowest $EWF_{b,ir}$ (1.29 $m^3$ $USD^{-1}$ in 2016). The difference between $EWF_{g,ir}$ and $EWF_{g,rf}$ of spring wheat was
the largest, which were 3.11 $m^3$ $USD^{-1}$ and 6.94 $m^3$ $USD^{-1}$, respectively, in 2016. The difference between $EWF_{g,ir}$ and $EWF_{g,rf}$
of tobacco was the smallest, which were 0.59 $m^3$ $USD^{-1}$ and 0.83 $m^3$ $USD^{-1}$, respectively, in 2016. Table 2 also lists the annual
blue and green CWU and yield under irrigated and rainfed conditions by crops in China for the years 2001 and 2016. It can be
seen that for all the crops, $CWU_{g,ir}$ was 21% (sugarcane) -55% (spring wheat) smaller than $CWU_{g,rf}$ in 2016. Therefore, it is
possible to result in $EWF_{g,ir}$ being much smaller than $EWF_{g,rf}$. During the study period, the $EWF_{g,rf}$ of cash crops decreased
most significantly. As for the $EWF_{b,ir}$, the downward trend of cash crops was more significant, compared to that of grain crops.
The M-K test results in Table 3 further confirmed the above results, as the M-K statistical values of all crops' EWF passed the
significance level test of $p<0.05$. M-K test results for the EWF of most crops were at the similar significance level as for the
corresponding PWF. It is mainly because the M-K test results of the prices of most crops were at the same significant level as
the corresponding M-K test results of the yields. Due to the significantly increased price, the EWF M-K test result of soybean
was -2.116, which was higher than the test result of corresponding PWF (-0.675). Cotton is another crop worthy of attention.
M-K test result for EWF of cotton was -2.476, whose significance level was lower than that of PWF. This is mainly due to
fluctuations in the price of cotton. In addition, it can be seen that the changes of $EWF_{b,ir}$ of most crops were not as obvious as
those of $EWF_{g,ir}$ and $EWF_{g,ir}$. It indicates that there is more potential in optimizing the economic benefit of agricultural blue
water input.
Figure 6a and 6b show the spatial distribution of EWF of grain and cash crops, respectively. Generally, the EWF of grain crop
was higher in Inner Mongolia and north-western China (Shaanxi, Gansu, Ningxia and Xinjiang); Guangdong, Jiangxi, Fujian,
Zhejiang and other south-eastern coastal provinces were at a relatively low level. The northwest, with higher PWF, has lower
crop prices due to the relatively underdeveloped economies. In contrast, the economically advanced southeast coastal
provinces have both low crop water consumption and higher prices. And the EWF of the 31 provinces showed a significant
declining trend over time, which was consistent with the characteristics of PWF of grain crop above (Fig. 3a). Specifically,
Gansu province with the highest PWF of grain crop in north-western China (mean 1.43 $m^3$ $kg^{-1}$) also had the highest EWF in
the top three (mean 8.34 $m^3$ $USD^{-1}$), with a significant decline of 46% over time, from 13.28 $m^3$ $USD^{-1}$ in 2001 to 7.12 $m^3$
$USD^{-1}$ in 2016. Another high value area in the northwest is Shaanxi, where winter wheat and spring maize were the main grain
crops (44% and 47% of all grain crops, respectively in 2016). The EWF and PWF in Shaanxi (mean 8.15 $m^3$ $USD^{-1}$ and 1.39
$m^3$ $kg^{-1}$) were second only to those in Gansu. In contrast, the EWF and PWF (mean 4.49 $m^3$ $USD^{-1}$ and 0.94 $m^3$ $kg^{-1}$) in Fujian,
with rice as the main grain crop (86% of all grain crops in 2016) were far lower than the national average (mean 5.41 $m^3$ $USD^{-1}$
and 1.01 $m^3$ $kg^{-1}$).
Concerning the composition of blue and green water for grain crop, the $EWF_{b,ir}$ in north-western China was lower, while the
$EWF_{g,ir}$ and $EWF_{g,rf}$ were higher. In contrast, the $EWF_{b,ir}$ in southern China was higher, while the $EWF_{g,ir}$ and $EWF_{g,rf}$
were lower. Specifically, in the northwest region, Ningxia had the highest $EWF_{g,ir}$ and $EWF_{g,rf}$ (mean 5.25 $m^3$ $USD^{-1}$ and 8.35
$m^3$ $USD^{-1}$, respectively), while the $EWF_{b,ir}$ was only 7.28 $m^3$ $USD^{-1}$, far lower than the national average (15.49 $m^3$ $USD^{-1}$).
Instead, the $EWF_{g,ir}$ and $EWF_{g,rf}$ in Yunnan were close to the national average level (3.59 $m^3$ $USD^{-1}$ and 5.31 $m^3$ $USD^{-1}$), and
$EWF_{b,ir}$ was the highest (52.05 $m^3$ $USD^{-1}$). This is mainly because Yunnan is located in the southwest, where the climate is
humid and rainfall is abundant. The yields of maize and rice mainly planted are basically guaranteed under the condition of
natural rainfall, with an extremely limited increase brought by irrigation. The EWF of cash crop had no obvious spatial
clustered phenomenon, decreasing significantly over time in 31 provinces, which was consistent with the spatial evolution
characteristics of the corresponding PWF previously discussed (Fig. 3b).

Figure 6. Temporal and spatial evolution of economic value-based water footprint (EWF) of (a) grain and (b) cash crops in
China.

Table 4 shows the global Moran's I of EWF of grain and cash crops. The average Moran's I of EWF of grain crop (0.482) was
higher than the PWF (0.263). Spatial agglomeration existed in most provinces, which was more stable over time. Differently
from grain crop, the spatial pattern of EWF of cash crop did not show obvious agglomeration, with average Moran's I of 0.016.
The LISA cluster maps of EWF of grain and cash crops are shown in Fig. 7. The H-H regions of EWF for grain crop were
mainly concentrated in Ningxia, Gansu, Shaanxi, Shanxi, Inner Mongolia, and L-L regions were mainly concentrated in
Guangdong, Zhejiang, Fujian, Jiangxi. During the research period, the EWF of grain crop showed an obvious and stable
positive spatial correlation. Generally, the spatial agglomeration pattern of EWF of grain crop was stable. As for cash crop,
the LISA maps of four representative years shows great changes. Only in 2011, it shows a certain positive spatial correlation,
with 4 provinces (Hunan, Hubei, Chongqing and Guizhou) in H-H regions. Overall, the EWF of cash crop did not show obvious
spatial agglomeration. For a same crop, the spatial variations of its PWF are defined by climate and productivity. The price is
one of the main factors defining the EWF. While in related to the cluster maps shown in the current results for grain and cash
crops, the main factor is the cultivation distribution. Regarding the grain crops, the cultivation distributions of major grain
crops in China show obvious spatial agglomeration characteristics. For instance, rice is mainly distributed in central and
southern China (Hubei, Hunan, Jiangxi, Guangdong and Guangxi). Winter wheat is concentrated in Huang-Huai-Hai Plain
(Shandong, Henan, Jiangsu, Anhui and Hebei). Whereas regarding the cash crops, the dominant crop differs among provinces
(see Fig. 10b) which resulted in obvious scattered characteristics in related WFs. For example, in the northwest regions, there
is only Xinjiang where cotton is planted on a large scale, and almost no cotton is planted in the surrounding provinces. In
addition, crop prices in the main producing provinces are generally lower, while vary affected by the regional economic level.
For example, both Henan and Shandong are the main producing areas of winter wheat, but the price (0.21 USD kg$^{-1}$ in 2016)
in Shandong, which has a more developed economy, was higher than that in Henan (0.17 USD kg$^{-1}$).

Figure 7. The LISA cluster maps of economic value-based water footprint (EWF) of (a) grain and (b) cash crops.

**3.3 Synergy evaluation of PWF and EWF**
Figure 8a and 8b show the SI between PWF and EWF of grain and cash crops across 31 provinces, respectively over years.
Concerning grain crop, the number of provinces with negative SI were increasing. Over time, the areas with negative SI
gradually expanded to the south. The SI was mostly negative in Beijing-Tianjin-Hebei, Inner Mongolia and north-western
China. In 2016, the SI of Shaanxi was -1.13, the lowest in China. The SI of Jiangxi, Chongqing, Hubei, Hunan, Jiangsu,
Zhejiang, Shanghai, and other coastal areas in south-eastern China was positive. In 2016, the SI of Jiangxi was 0.62, the highest
in China. Overall, the SI of grain crop was negative in Inner Mongolia and north-western China (Shaanxi, Gansu, Ningxia),
whereas in Guangdong, Jiangxi, Fujian, Zhejiang and other coastal areas in south-eastern China it was positive, with a clustered
distribution. With the development of water-saving technologies and the improvement of agricultural management, China has
made gratifying progress in the efficient use of water for crop production from a single physical or economic perspective.
However, only by combining the physical and economic perspectives can we gain a deeper understanding of the underlying
problems and catch the synergies, trade-offs and even lose-lose relationships between reducing the water resources input for
harvesting crop yields and optimizing the economic benefits per unit of water consumption in different regions.
As for cash crop, the SI of Tianjin, Jiangxi and Hunan was always negative, and the lowest in China (multi-year mean values
-0.98, -0.90 and -0.74, respectively). Overall, there were more provinces with negative SI of cash crop, and the incongruity
between PWF and EWF of cash crop was more significant than that of grain crop. Interestingly, the provinces with the most
severe negative SI for grain crops had positive SI for cash crops. The highest SI of cash crop in 2016 occurred in Shanghai
(0.39), which was lower than the SI of grain crop in the same year (0.45). At the same time, the SI of grain and cash crops in
Tianjin, Tibet and Xinjiang decreased significantly. In more provinces, the SI of grain and cash crop varied greatly and was
not synchronised. For example, the SI of grain crop in Inner Mongolia and Fujian increased significantly, while the SI of cash
crop showed a downward trend. Furthermore, the SI of cash crop in Shaanxi and Gansu increased significantly, while the SI
of grain crop did not change significantly.

Figure 8. Temporal and spatial evolution of synergy evaluation index (SI) of (a) grain and (b) cash crops.

Taking 2016 as an example, we further look at the reasons for the "lose-lose" relationship between reducing the water resources
input for harvesting crop yields and optimizing the economic benefits per unit of water consumption in both grain and cash
crops (see Fig. 9), from the perspective of planting structure (see Fig. 10). Shaanxi province had the highest PWF in China
(1.23 $m^3$ $kg^{-1}$), and the second highest EWF (7.48 $m^3$ $USD^{-1}$). In Shaanxi, winter wheat and spring maize with high water
consumption and low yield accounted for more than 90% of the total sown area of grain crops, with yields lower than the
national averages by 24% and 26%, respectively. Moreover, the price of wheat in Shaanxi province (0.17 USD $kg^{-1}$) was lower
than the national average (0.19 USD $kg^{-1}$). The reasons for high water consumption per unit of grain production coupled with
poor economic benefits in Shaanxi province can be attributed to the above two points. In contrast, in Jiangxi province, where
rice, which has low water consumption intensity, is the main grain crop (rice accounting for 95% of the grain crops), PWF and
EWF were 0.77 $m^3$ $kg^{-1}$ and 3.63 $m^3$ $USD^{-1}$, well below the national averages (0.93 $m^3$ $kg^{-1}$, 5.04 $m^3$ $USD^{-1}$).
As for cash crop, the PWF of Tianjin was 1.92 $m^3$ $kg^{-1}$, the highest in China, and the EWF was 3.26 $m^3$ $USD^{-1}$, the fifth highest
in China, which was significantly higher than the national average (2.05 $m^3$ $USD^{-1}$). It can be seen from Fig. 10b that cotton
accounted for the largest proportion (70%) in the planting structure of cash crops in Tianjin. Cotton consumed the most water
per yield unit of cash crops, while the price unit of cotton in Tianjin was the second lowest in China (1.11 USD $kg^{-1}$), which
did not reflect the advantage of cotton as a high-value crop. Jiangxi province showed the highest EWF in China (3.86 $m^3$ $USD^{-1}$
$^{1}$), and a PWF (0.96 $m^3$ $kg^{-1}$) which was also higher than the national average (0.46 $m^3$ $kg^{-1}$). Figure 10b shows that citrus
(planting area accounting for 29% of cash crops) and rapeseed (planting area accounting for 48% of cash crops) are the main
cash crops in Jiangxi. However, the price unit of citrus in Jiangxi was the third lowest (0.17 USD kg$^{-1}$, only 62% of the national
average), and the yield of rapeseed was also the third lowest (1.34 t ha$^{-1}$, 32% lower than the national average). In contrast, the
main cash crop in Shanxi was apple (planting area accounting for 87% of cash crops), with low water consumption intensity
and a yield which was the second highest in China (28.5 t ha$^{-1}$), 1.5 times larger than the national average (18.9 t ha$^{-1}$).

Figure 9. Production-based water footprint (PWF) versus economic value-based water footprint (EWF) of (a) grain and (b)
cash crops per province in 2016.

Figure 10. Planting structure of (a) grain and (b) cash crops in 31 provinces in 2016.

**4 Discussion**

The goal of WF regulation is to reduce its magnitude to a sustainable level (Hoekstra, 2013), but the challenges faced during
implementing sustainable development are rarely encountered in a single dimension. However, previous research has most
commonly adopted a single perspective approach to WF analysis. Based on the temporal and spatial evolution of PWF and
EWF, the synergy evaluation index (SI) is constructed to achieve a more comprehensive assessment in this study. This
approach has led to some differences in the results of WF compared to previous research.
Table 5 compares the PWF results of crops production between the current study and previous ones. Differently from
Mekonnen and Hoekstra (2011) and Zhuo et al. (2016b), this study distinguishes between wheat and maize varieties when
calculating the WF, despite China's wheat production is mainly of winter wheat (accounting for 95% in 2016). Due to the
differences of varieties, water consumption intensity and planting conditions, it is necessary to distinguish between crops in
the provinces where spring wheat is the main crop. In addition, due to the differences in model selection and parameters, the
calculation results will also be different. For example, Mekonnen and Hoekstra (2011) used CROPWAT model and checked
the crop yield at the national scale, while this study chooses AquaCrop model and checks the crop yield at the provincial level.
Both the studies of Mekonnen and Hoekstra (2011) and Zhuo et al. (2016b) were based on the 5 arc-minute grid, while this
research calculates the WF based on the meteorological station scale. In general, however, the crop production WF in this
study is close to that of previous studies, which shows the rationality of the calculated results.
Table 6 compares the EWF of this study with previously calculated results of the economic water productivity. There were no
existing EWF values for China's cases. We wish to show the available values on EWF of crops, while for countries other than
China. Since the economic water productivity is numerically equal to the reciprocal of the EWF, the previous results are
expressed in the form of EWF for comparison. The results for wheat production show that, although the average EWF is close,
differences in crop varieties, planting environment, and climate condition result in huge differences in $EWF_{b,ir}$ under the same
production mode. Therefore, specific problems should be investigated separately. Selection and adjustment of production
mode should be made according to local conditions to promote coordinated development.
From the results of the multi-perspective analysis conducted in this study, we found that with the increase of yield and price,
the PWF and EWF of crop production both showed a decreasing trend, and the EWF decreased more significantly compared
with the PWF. The change of WF of cash crops was more obvious than that of grain crops. In terms of the spatial pattern,
compared with cash crops, WF of grain crops had a more significant spatial correlation, and the spatial distribution of PWF
was similar to that of EWF. H-H areas mainly gathered in north-western China, while L-L areas in south-eastern coastal
provinces. The average Moran's I of EWF (0.482) was higher than that of PWF (0.263).
Moreover, results show that as for grain production at the national level, the $EWF_{b,ir}$ (mean 15.49 $m^3$ $USD^{-1}$) was much higher
than the $EWF_{g,ir}$ (mean 3.11 $m^3$ $USD^{-1}$), and the $EWF_{g,rf}$ (mean 5.31 $m^3$ $USD^{-1}$) was the closest to the average EWF in irrigation
and rainfed agriculture (mean 5.41 $m^3$ $USD^{-1}$). Compared with grain crops, the difference between $EWF_{g,ir}$ and $EWF_{g,rf}$ of cash
crops was smaller, with average values of 1.90 $m^3$ $USD^{-1}$ and 2.48 $m^3$ $USD^{-1}$, respectively. Moreover, the EWF of cash crops
was lower than that of grain crops. It was more cost-effective to increase the input of green water than that of blue water during
crop production. In north-western China, the $EWF_{b,ir}$ was lower, while the $EWF_{g,ir}$ and $EWF_{g,rf}$ were higher; on the contrary, in
southern China, the $EWF_{b,ir}$ was higher, while the $EWF_{g,ir}$ and $EWF_{g,rf}$ were lower. Therefore, the utilisation efficiency of green
water resources should be improved through water retention by tillage system and mulching. Meanwhile, more blue water can
be generated through rainwater harvesting (Hoekstra, 2019). Specifically, we suggest two measures to increase the blue water
efficiency in northern China. One is the rainwater harvesting in rainy season, especially for the short-time heavy rain which
cannot effectively used by crops but easily cause soil erosion. The other one is reducing blue water consumption and loss at
field by popularizing water-saving irrigation techniques and mulching practices. Such measure is helpful to improve the
utilisation efficiency of both blue and green water.  Based on the current results, we recommend the government to improve
agricultural water use efficiency through the extension of water-saving irrigation techniques and better agricultural inputs
management, especially in northwest China. High water consumption and low economic value crops' acreages in non-primary
production areas should be reduced. For the southern regions with abundant rainwater resources, the economic benefits of
irrigation are very limited, on the contrary, rainfed agriculture has obvious advantages and the potential to increase economic
benefits. Therefore, farmers should improve the water conservation rate and the utilization efficiency of green water through
farming system and coverage to reduce the amount of water used for irrigation. The government should also give financial
subsidies for agricultural production to those provinces where there were lose-lose relationships between reducing the water
resources input for harvesting crop yields and optimizing the economic benefits per unit of water consumption. Finally,
improve the field managements especially in utilization rate of chemical fertilizers and pesticides to increase agricultural
productivity further (Zhang et al., 2013).
There was a serious lose-lose relationship between reducing the water resources input for harvesting crop yields and optimizing
the economic benefits per unit of water consumption both in grain and cash crops. In terms of grain production, the water
consumption per yield was large, but the economic benefit per water consumption unit was poor in the northwest region, while
the opposite was true in the southeast coastal region. Over time, the lose-lose relationship has not been alleviated, showing
a relatively stable spatial pattern. Through analysis, this study shows that the unreasonable regional planting structure and crop
price may be the direct cause of the incongruity between water resource consumption and economic value creation for crop
production in China.
The study reveals the synergies and trade-offs of crop PWFs and EWFs. However, it is undeniable that there are some
limitations and shortcomings. Firstly, in the calculation of WF, although the accuracy of AquaCrop model in simulating crop
water consumption and yield, soil field water, and fertiliser management types under different climatic conditions has been
widely demonstrated, the uncertainty of results caused by the uncertainty of input parameters must be acknowledged (Zhuo et
al., 2014). Secondly, this paper does not make a specific distinction between crop irrigation methods. In fact, the difference of
WF results caused by different irrigation methods cannot be ignored (Wang et al., 2019). Thirdly, when calculating the WF, it
is assumed that the change of crop irrigation and rainfed planting area only occurs in the data grid based on 2000, and the
migration of crop harvesting zone is not considered. Finally, this study does not focus specifically on the effects of field water
and fertiliser management measures. Although there are restrictions on the availability of crop price unit data in the selection
of research objects, it is still representative because the crops selected in this paper accounts for more than 85% of the national
crop production. As for the study perspective, this article focuses on trade-offs between water consumption and economic
value creation in crop production. In fact, the ecological impacts on the environment cannot be ignored. Therefore, further
research is expected to tackle this limitation by including the ecological impacts on the environment in a more comprehensive
assessment. In addition, it should be noted that in the current study, the SI measures, considering the spatial heterogeneities in
crop WFs among provinces, the synergy levels between the current PWF and EWF. The synergy (both the PWF and EWF are
lower than the national averages), trade-off (one is higher than the national average while the other is lower), or lose-lose (both
are higher than the national averages) situation can be identified. The most optimized situation means high economic value
generated by low water consumption. For the two provinces with high SI values, they were both in an advantageous position,
while the one with a higher SI values performed better in terms of synergy between PWF and EWF. If the reference value is
set by the WF benchmark (Mekonnen and Hoekstra, 2014b), then the SI will show information on efficiency. The meaning is
totally different from the current one. Choosing proper reference value for different functions is highly recommended.

Table 5. Comparison between production-based water footprint (PWF) of crops production in mainland China in the current study and previous studies.

Table 6. Comparison between economic value-based water footprint (EWF) in the current results and previous studies.

## 5 Conclusions

Based on temporal and spatial evolution analysis of WF of China's crop production from a physical and economical perspective, this study makes a comprehensive assessment by constructing a SI between PWF and EWF, and reveals the synergies and trade-offs of crop water productivity and its economic value. Results show that:

(1) With the increase of yield unit and price unit, the PWF and EWF of crop production both showed a decreasing trend, and the EWF decreased more significantly. The change of WF of cash crops was more obvious than that of grain crops.

(2) Compared to cash crops, WF of grain crops had a more significant spatial correlation, and the spatial distribution of PWF was similar to that of EWF. H-H areas mainly gathered in north-western China, while L-L areas in southeast coastal provinces. The average Moran's I of EWF (0.482) was higher than that of PWF (0.263).

(3) The economic benefits of blue water and green water differed greatly, and the difference showed to be more significant for grain crop than for cash crop. Moreover, the EWF of cash crops was lower than that of grain crops. It was found to be more cost-effective to increase the input of green water than that of blue water during crop production.

(4) In terms of grain production, the water consumption per yield unit was large but the economic benefit per water consumption unit was poor in the northwest region, while the opposite was true in the southeast coastal region. The trade-offs have not been alleviated over time, showing a relatively stable spatial pattern. These findings show that the unreasonable regional planting structure and crop price may be the direct cause of the lose-lose relationships between water resource consumption and economic value creation for crop production, so this issue should be tackled by coordinated governmental action, to balance the economic benefits of the water-intensive crops in different regions.

## Data availability

Data sources of carrying out the study are listed in the section 2.6 Data sources. Data generated in this paper is available by contacting L Zhuo.

## Author contributions

La Zhuo and Xi Yang designed the study. Xi Yang carried it out. Xi Yang prepared the manuscript with contributions from all co-authors.

## Competing interests

The authors declare that they have no conflict of interest.

## Acknowledgements

This work was supported by the National Key Research and Development Plan [2018YFF0215702], the National Natural Science Foundation of China Grants [51809215], the Fundamental Research Funds for the Central Universities [2452017181], and the 111 Project [B12007].

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

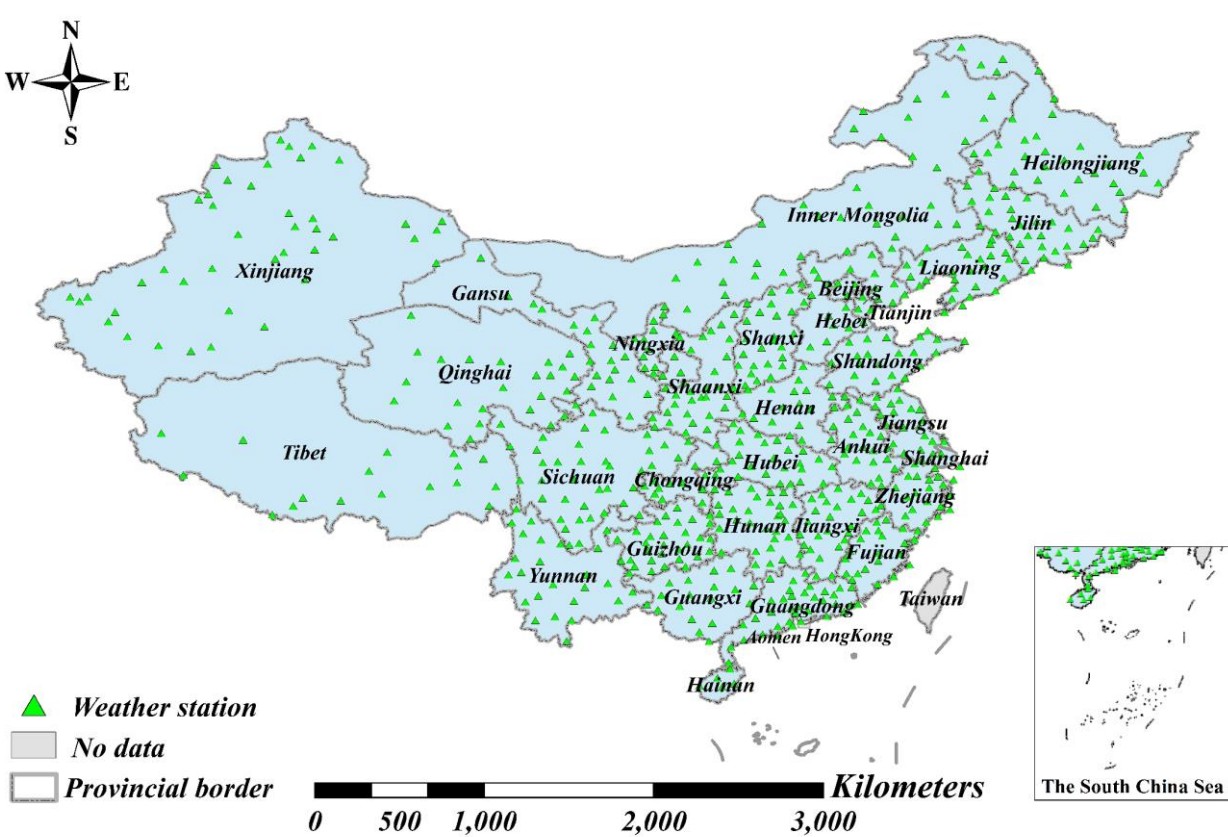


**Figure 1: Considered weather stations across mainland China.**

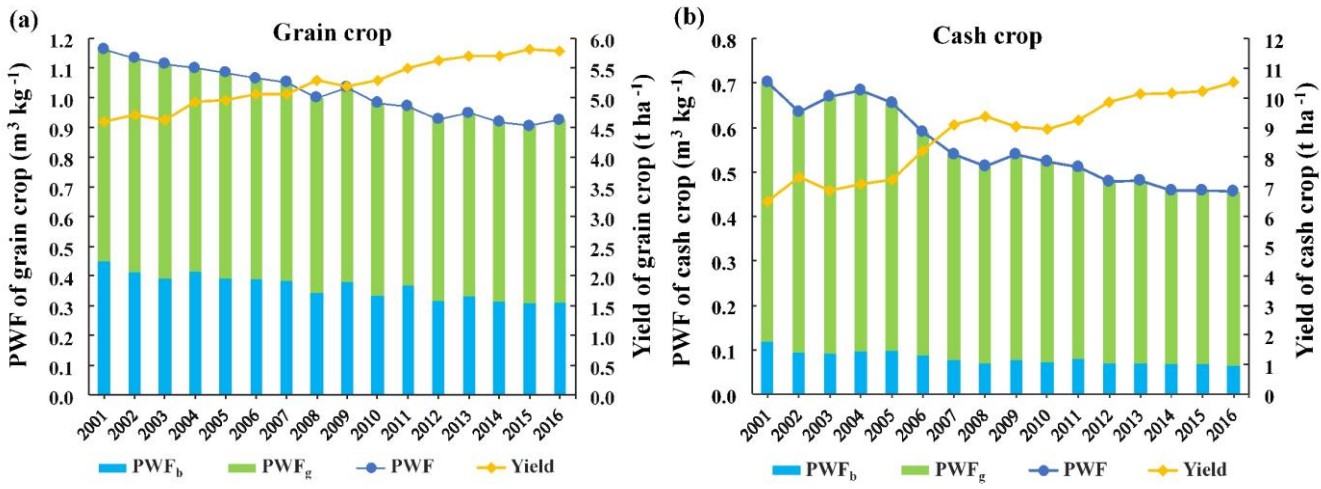


Figure 2: Interannual variability of national average production-based water footprint (PWF) of (a) grain and (b) cash crops in China over 2001-2016.

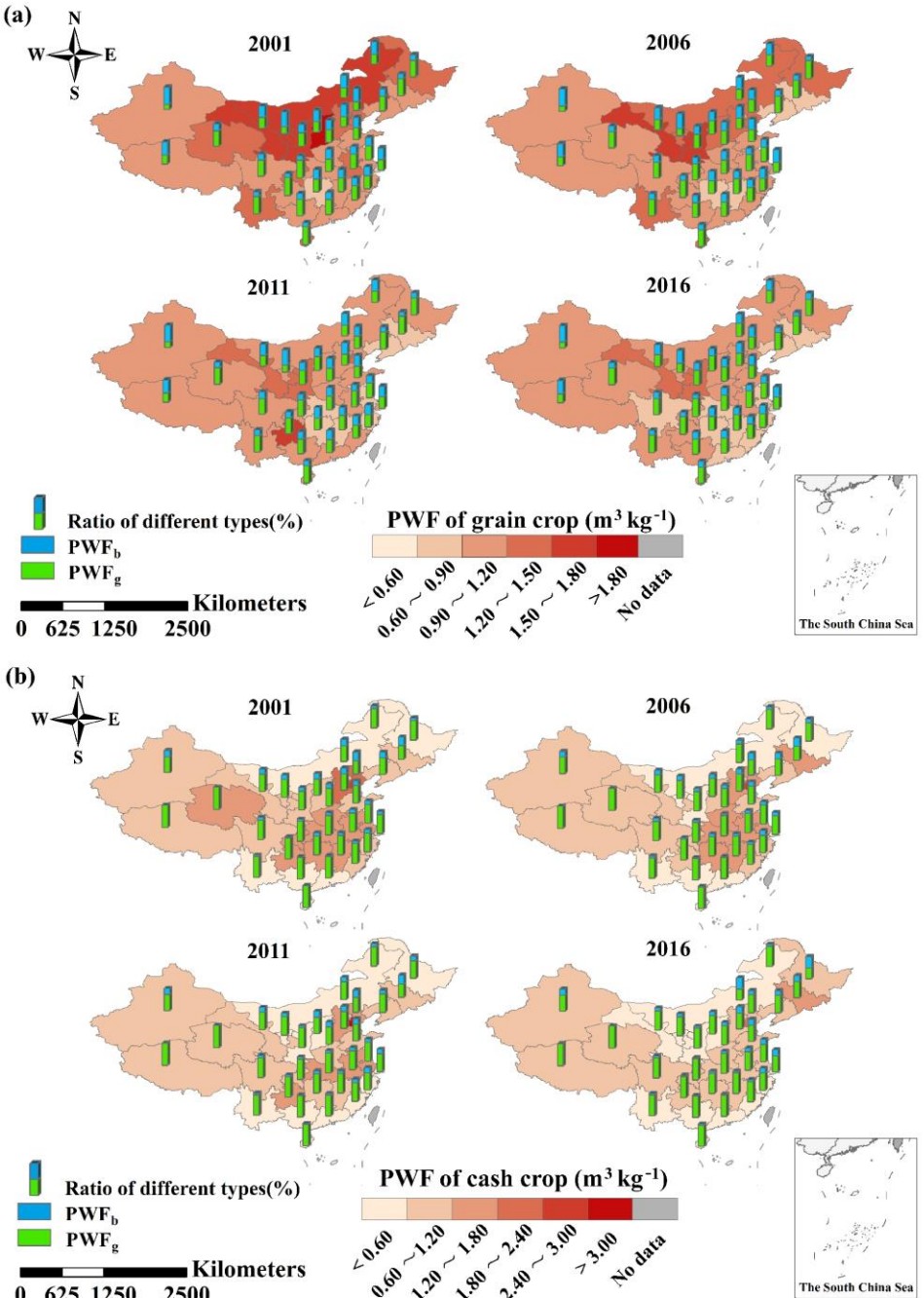


Figure 3: Temporal and spatial evolution of production-based water footprint (PWF) of (a) grain and (b) cash crops in China.

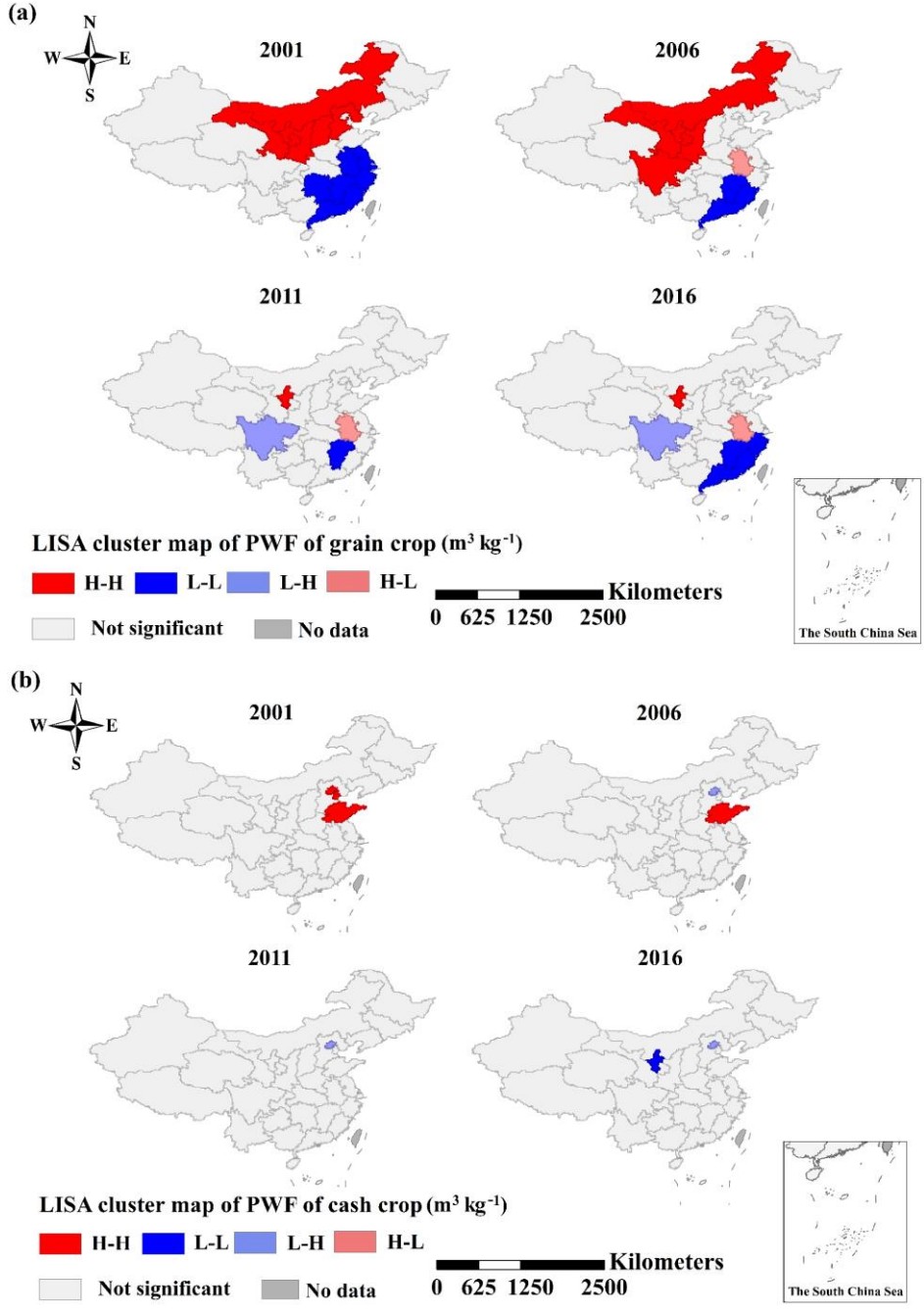


Figure 4:The LISA cluster maps of production-based water footprint (PWF) of (a) grain and (b) cash crops.

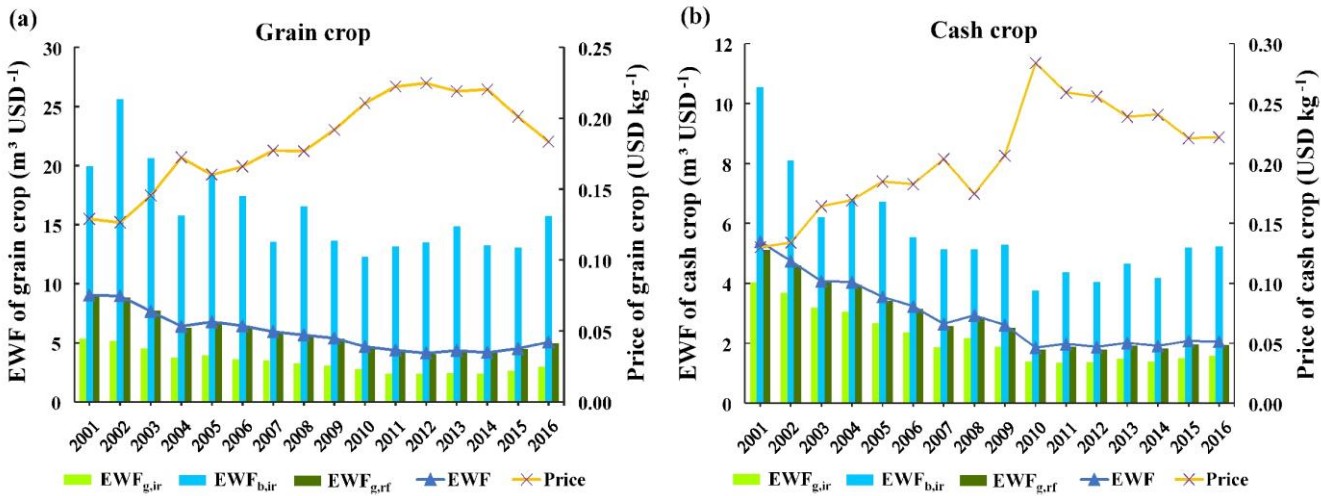


**Figure 5: Interannual variability of economic value-based water footprint (EWF) of (a) grain and (b) cash crops in China over 2001-**
**2016.**

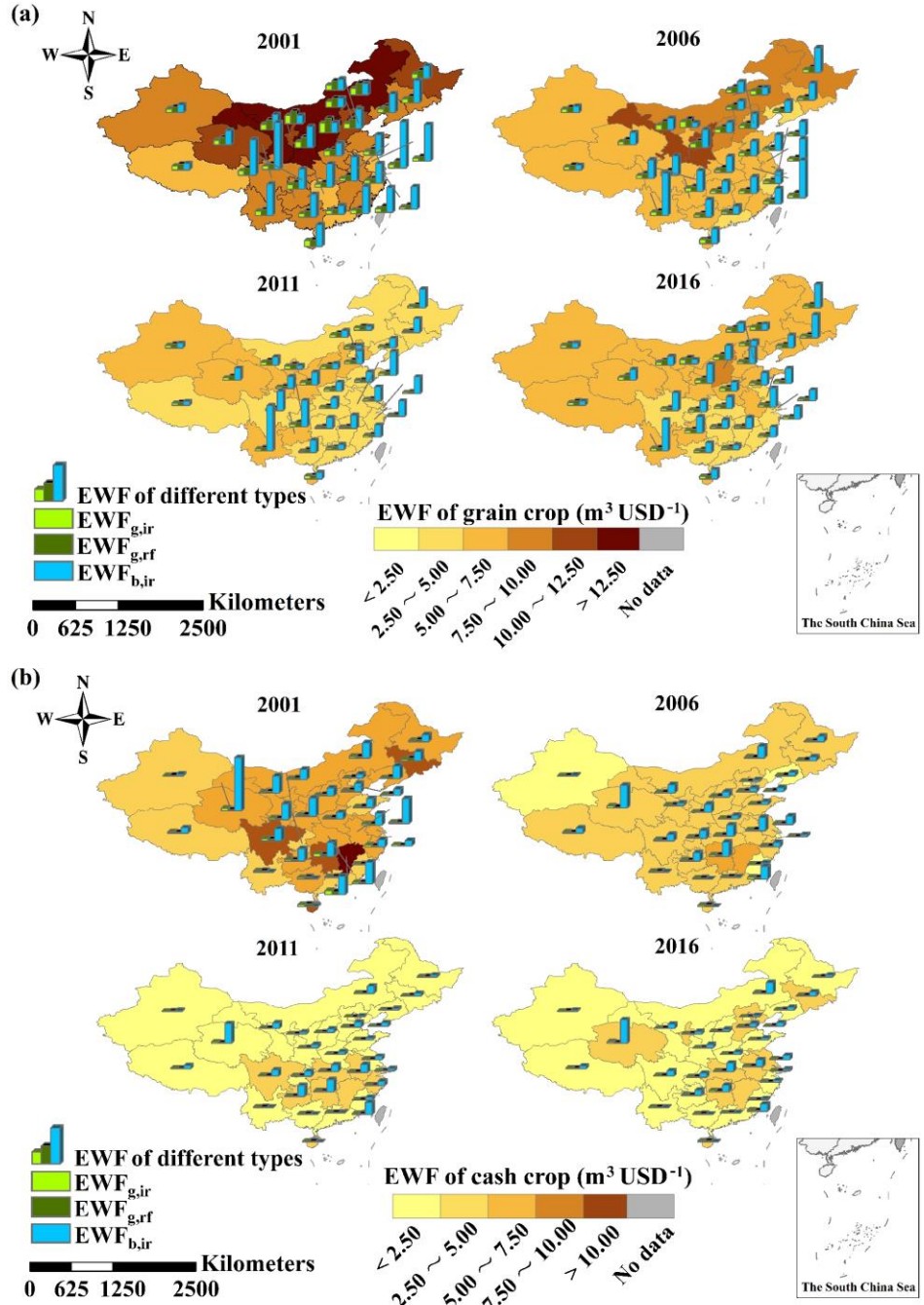


Figure 6: Temporal and spatial evolution of economic value-based water footprint (EWF) of (a) grain and (b) cash crops in China.

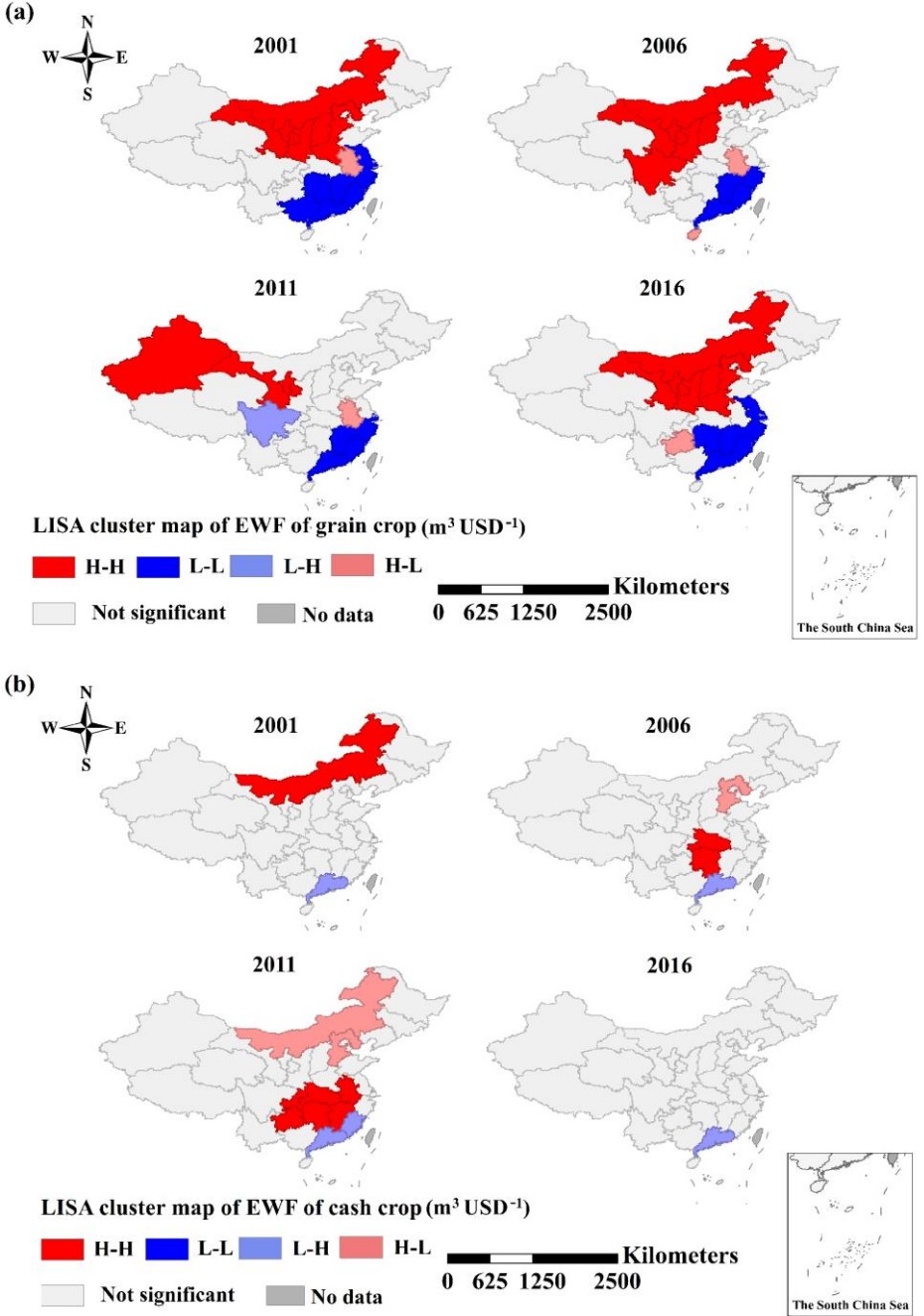


**Figure 7: The LISA cluster maps of economic value-based water footprint (EWF) of (a) grain and (b) cash crops.**

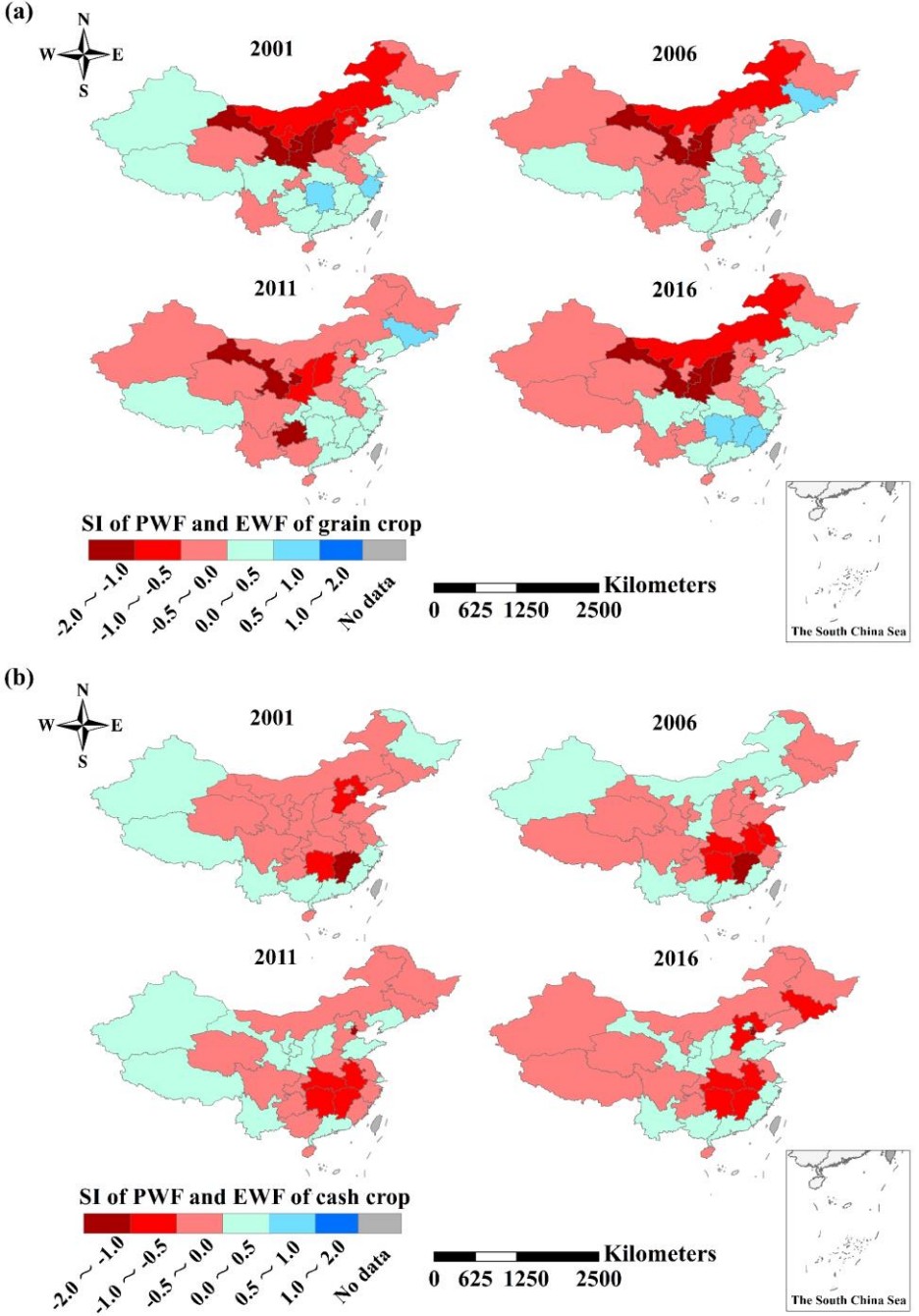


**Figure 8: Temporal and spatial evolution of synergy evaluation index (SI) of (a) grain and (b) cash crops.**

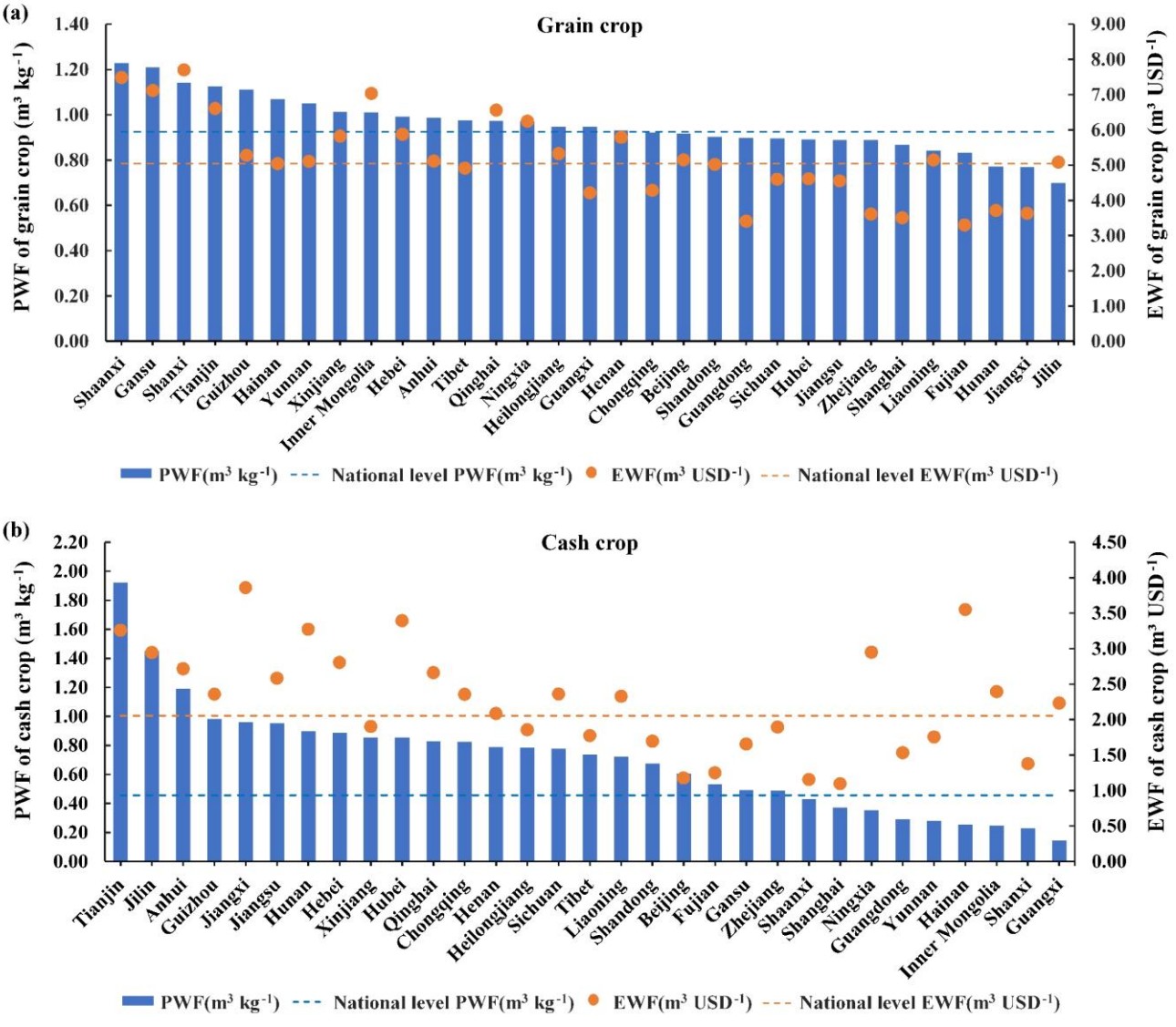


**Figure 9: Production-based water footprint (PWF) versus economic value-based water footprint (EWF) of (a) grain and (b) cash**
**crops per province in 2016.**

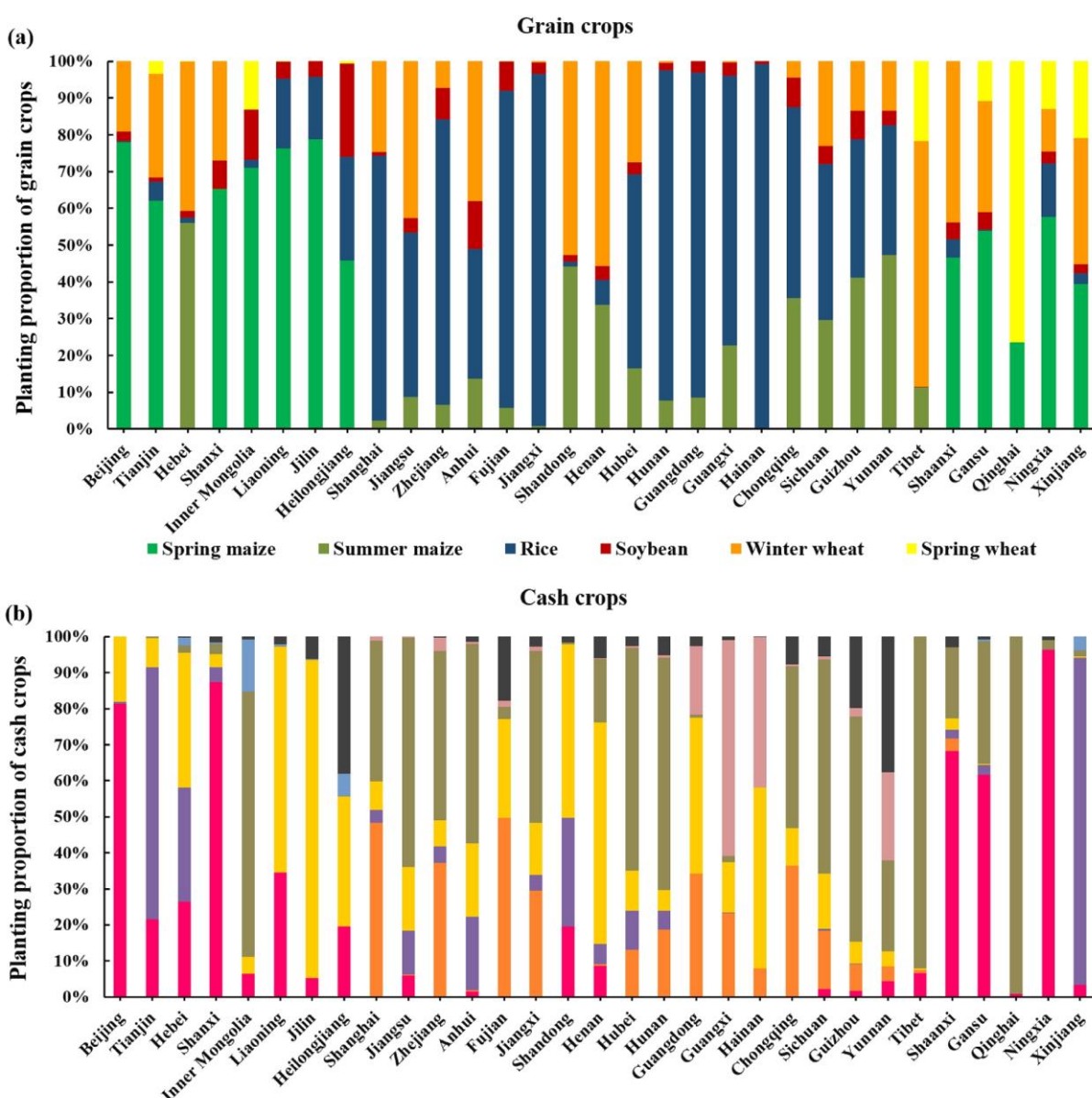

**Figure 10: Planting structure of (a) grain and (b) cash crops in 31 provinces in 2016.**

**Table 1. Number of meteorological stations per province.**

| Region | Province | Number of weather stations | Climatic zone |
|---|---|---|---|
| North-central | Beijing | 3 | Temperate |
| | Tianjin | 3 | |
| | Shanxi | 27 | |
| Northeast | Inner Mongolia | 36 | Continental temperate & temperate |
| | Liaoning | 25 | |
| | Jilin | 29 | |
| | Heilongjiang | 34 | |
| Huang-Huai-Hai | Hebei | 19 | Temperate |
| | Shandong | 21 | |
| | Henan | 17 | |
| | Anhui | 21 | |
| Southeast | Shanghai | 1 | Sub-tropics |
| | Zhejiang | 21 | |
| | Fujian | 22 | |
| Yangtze (middle & lower reaches) | Jiangsu | 22 | Sub-tropics |
| | Jiangxi | 26 | |
| | Hubei | 27 | |
| | Hunan | 29 | |
| South-central | Guangdong | 36 | Sub-tropics & tropics |
| | Guangxi | 18 | |
| | Hainan | 5 | |
| Southwest | Chongqing | 11 | Sub-tropics |
| | Sichuan | 38 | |
| | Guizhou | 31 | |
| | Yunnan | 25 | |
| | Tibet | 17 | |
| Northwest | Shaanxi | 32 | Continental temperate & plateau and mountain |
| | Gansu | 23 | |
| | Qinghai | 25 | |
| | Ningxia | 12 | |
| | Xinjiang | 42 | |



Table 2. National average production-based water footprint (PWF) and economic value-based water footprint (EWF) of crops in China for the years 2001 and 2016.

| Crop | Year | Irrigated | | | | | | Rainfed | | | | Price | Yield | PWF$_b$ | PWF$_g$ | PWF | EWF |
|------|------|-----------|--|--|--|--|--|---------|--|--|--|-------|-------|---------|---------|-----|-----|
| | | CWU$_{b,ir}$ | CWU$_{g,ir}$ | PWF$_{ir}$ | EWF$_{b,ir}$ | EWF$_{g,ir}$ | Y$_{IR}$ | CWU$_{g,rf}$ | PWF$_{rf}$ | EWF$_{g,rf}$ | Y$_{RF}$ | | | | | | |
| | | m³ha⁻¹ | m³ha⁻¹ | m³kg⁻¹ | m³USD⁻¹ | m³USD⁻¹ | kg ha⁻¹ | m³ha⁻¹ | m³kg⁻¹ | m³USD⁻¹ | kg ha⁻¹ | USD kg⁻¹ | kg ha⁻¹ | m³kg⁻¹ | m³kg⁻¹ | m³kg⁻¹ | m³USD⁻¹ |
| Grain crop | 2001 | 3266 | 2599 | 1.16 | 19.93 | 5.32 | 5059 | 4420 | 1.17 | 9.05 | 3788 | 0.13 | 4596 | 0.45 | 0.71 | 1.16 | 9.01 |
| | 2016 | 2995 | 2797 | 0.94 | 15.70 | 2.96 | 6193 | 4674 | 0.91 | 4.94 | 5153 | 0.18 | 5780 | 0.32 | 0.61 | 0.93 | 5.04 |
| Winter wheat | 2001 | 3590 | 2617 | 1.47 | 22.95 | 6.91 | 4220 | 4428 | 1.48 | 11.68 | 2987 | 0.13 | 3806 | 0.62 | 0.85 | 1.47 | 11.62 |
| | 2016 | 3329 | 2776 | 1.05 | 15.02 | 3.19 | 5819 | 4726 | 1.02 | 5.42 | 4639 | 0.19 | 5402 | 0.40 | 0.64 | 1.04 | 5.54 |
| Spring wheat | 2001 | | | | | | | | | | | | | | | | |
| | 2016 | 4900 | 1750 | 1.52 | 18.88 | 3.11 | 4373 | 3898 | 1.30 | 6.94 | 2992 | 0.19 | 4237 | 1.05 | 0.46 | 1.51 | 8.02 |
| Spring maize | 2001 | 4683 | 2279 | 1.19 | 18.57 | 5.27 | 5860 | 4268 | 1.15 | 9.87 | 3701 | 0.12 | 4666 | 0.45 | 0.72 | 1.17 | 10.04 |
| | 2016 | 3943 | 2557 | 0.86 | 16.96 | 3.41 | 7586 | 4633 | 0.80 | 6.18 | 5791 | 0.13 | 6435 | 0.22 | 0.60 | 0.82 | 6.36 |
| Summer maize | 2001 | 2822 | 2844 | 1.13 | 36.18 | 5.58 | 5030 | 4681 | 1.07 | 9.19 | 4362 | 0.12 | 4725 | 0.32 | 0.78 | 1.10 | 9.45 |
| | 2016 | 2564 | 2983 | 0.99 | 64.25 | 4.35 | 5605 | 4773 | 0.90 | 6.96 | 5297 | 0.13 | 5439 | 0.21 | 0.73 | 0.94 | 7.28 |
| Rice | 2001 | 2868 | 2568 | 0.86 | 23.78 | 3.71 | 6312 | 4324 | 0.80 | 6.25 | 5375 | 0.13 | 6163 | 0.39 | 0.46 | 0.85 | 6.63 |
| | 2016 | 2583 | 2876 | 0.79 | 20.79 | 1.96 | 6940 | 4563 | 0.71 | 3.11 | 6398 | 0.23 | 6862 | 0.33 | 0.45 | 0.78 | 3.39 |
| Soybean | 2001 | 3511 | 2677 | 2.84 | 19.31 | 8.15 | 2183 | 4378 | 3.12 | 13.32 | 1405 | 0.23 | 1625 | 0.61 | 2.40 | 3.01 | 12.87 |
| | 2016 | 2928 | 2779 | 2.80 | 24.43 | 5.21 | 2040 | 4627 | 2.78 | 8.68 | 1666 | 0.32 | 1796 | 0.57 | 2.22 | 2.79 | 8.70 |
| Cash crop | 2001 | 4224 | 3106 | 0.81 | 10.54 | 4.01 | 9033 | 3955 | 0.66 | 5.11 | 5954 | 0.13 | 6512 | 0.12 | 0.58 | 0.70 | 5.39 |
| | 2016 | 3722 | 3469 | 0.55 | 5.22 | 1.57 | 13158 | 4268 | 0.43 | 1.94 | 9945 | 0.22 | 10526 | 0.07 | 0.39 | 0.46 | 2.05 |
| Ground-nut | 2001 | 3810 | 2614 | 2.07 | 41.27 | 3.40 | 3103 | 4399 | 1.59 | 5.72 | 2771 | 0.28 | 2888 | 0.46 | 1.31 | 1.77 | 6.37 |
| | 2016 | 3228 | 3075 | 1.70 | 69.28 | 1.57 | 3712 | 4997 | 1.38 | 2.55 | 3626 | 0.54 | 3657 | 0.31 | 1.18 | 1.49 | 2.76 |
| Rapeseed | 2001 | | | | | | | 2066 | 1.29 | 5.92 | 1597 | 0.22 | 1597 | | 1.29 | 1.29 | 5.92 |
| | 2016 | | | | | | | 2065 | 1.04 | 2.73 | 1984 | 0.38 | 1984 | | 1.04 | 1.04 | 2.73 |
| Cotton | 2001 | 5291 | 2868 | 6.21 | 20.35 | 3.05 | 1314 | 4987 | 4.85 | 5.30 | 1029 | 0.92 | 1107 | 1.31 | 3.98 | 5.29 | 5.78 |
| | 2016 | 5035 | 3093 | 4.69 | 17.62 | 1.68 | 1732 | 4501 | 3.00 | 2.45 | 1500 | 1.23 | 1584 | 1.16 | 2.52 | 3.68 | 3.00 |
| Sugarcane | 2001 | 4199 | 5367 | 0.09 | 3.47 | 4.35 | 106621 | 7357 | 0.14 | 5.97 | 53811 | 0.02 | 60625 | 0.01 | 0.12 | 0.13 | 5.50 |
| | 2016 | 3809 | 5805 | 0.07 | 1.29 | 2.04 | 139879 | 7315 | 0.11 | 2.58 | 68656 | 0.04 | 74550 | 0.01 | 0.09 | 0.10 | 2.43 |
| Sugar beet | 2001 | | | | | | | 3850 | 0.14 | 5.43 | 26764 | 0.03 | 26764 | | 0.14 | 0.14 | 5.43 |
| | 2016 | | | | | | | 3602 | 0.06 | 1.60 | 57547 | 0.04 | 57547 | | 0.06 | 0.06 | 1.60 |
| Apple | 2001 | 4826 | 3235 | 0.75 | 34.96 | 2.83 | 10704 | 5114 | 0.54 | 4.47 | 9551 | 0.12 | 9687 | 0.05 | 0.51 | 0.56 | 4.71 |
| | 2016 | 4186 | 3579 | 0.36 | 4.96 | 0.74 | 21677 | 5457 | 0.30 | 1.13 | 18454 | 0.26 | 18883 | 0.03 | 0.28 | 0.31 | 1.17 |
| Citrus | 2001 | 3535 | 5017 | 0.94 | 45.86 | 3.68 | 9077 | 7534 | 0.88 | 5.52 | 8591 | 0.16 | 8769 | 0.15 | 0.75 | 0.90 | 5.68 |
| | 2016 | 3004 | 5318 | 0.53 | 9.43 | 1.34 | 15613 | 7908 | 0.55 | 2.00 | 14451 | 0.27 | 14701 | 0.04 | 0.50 | 0.54 | 1.99 |
| Tobacco | 2001 | 2365 | 2564 | 2.57 | 12.92 | 1.66 | 1918 | 3579 | 2.09 | 2.32 | 1715 | 0.90 | 1754 | 0.26 | 1.93 | 2.19 | 2.43 |
| | 2016 | 2010 | 2741 | 2.08 | 4.69 | 0.59 | 2289 | 3832 | 1.83 | 0.83 | 2094 | 2.21 | 2141 | 0.22 | 1.67 | 1.89 | 0.86 |


**Table 3. M-K analysis of production-based water footprint (PWF) and economic value-based water footprint (EWF) of the 14 crops.**

| | | Price USD kg⁻¹ | Yield kg ha⁻¹ | $PWF_b$ m³ kg⁻¹ | $PWF_g$ m³ kg⁻¹ | PWF m³ kg⁻¹ | EWF m³ USD⁻¹ | $EWF_{b,ir}$ m³ USD⁻¹ | $EWF_{g,ir}$ m³ USD⁻¹ | $EWF_{g,rf}$ m³USD⁻¹ |
|---|---|---|---|---|---|---|---|---|---|---|
| **Grain** | $Z_c$ | 3.557 | 5.088 | -4.727 | -4.277 | -4.997 | -4.097 | -3.017 | -4.187 | -4.007 |
| **crop** | Signific | ** | ** | ** | ** | ** | ** | ** | ** | ** |
| **Winter** | $Z_c$ | 4.007 | 5.178 | -3.737 | -4.547 | -4.547 | -4.547 | -1.936 | -4.547 | -4.547 |
| **wheat** | Signific | ** | ** | ** | ** | ** | ** | | ** | ** |
| **Spring** | $Z_c$ | 4.007 | 4.457 | -0.135 | -3.107 | -2.476 | -2.746 | 0.045 | -2.926 | -2.746 |
| **wheat** | Signific | ** | ** | | ** | * | ** | | ** | ** |
| **Spring** | $Z_c$ | 3.107 | 3.647 | -4.097 | -2.476 | -4.097 | -3.647 | -2.386 | -3.377 | -3.647 |
| **maize** | Signific | ** | ** | ** | * | ** | ** | * | ** | ** |
| **Summer** | $Z_c$ | 3.107 | 4.277 | -3.647 | -3.197 | -3.287 | -3.377 | 0.495 | -3.647 | -3.467 |
| **maize** | Signific | ** | ** | ** | ** | ** | ** | | ** | ** |
| **Rice** | $Z_c$ | 3.647 | 4.637 | -3.107 | -3.017 | -3.377 | -4.367 | -0.315 | -3.827 | -4.277 |
| | Signific | ** | ** | ** | ** | ** | ** | | ** | ** |
| **Soybean** | $Z_c$ | 2.116 | 1.126 | 1.846 | -1.396 | -0.675 | -2.116 | 0.135 | -2.656 | -2.296 |
| | Signific | * | | | | | * | | ** | * |
| **Cash** | $Z_c$ | 3.287 | 4.547 | -4.007 | -4.547 | -4.637 | -3.737 | -3.017 | -3.647 | -3.737 |
| **crop** | Signific | ** | ** | ** | ** | ** | ** | ** | ** | ** |
| **Ground-** | $Z_c$ | 2.926 | 4.547 | -3.467 | -3.287 | -3.917 | -3.377 | 0.405 | -3.017 | -3.197 |
| **nut** | Signific | ** | ** | ** | ** | ** | ** | | ** | ** |
| **Rapeseed** | $Z_c$ | 3.197 | 4.097 | 2.386 | -2.476 | -2.476 | -3.377 | | | -3.377 |
| | Signific | ** | ** | * | * | * | ** | | | ** |
| **Cotton** | Zc | 0.135 | 4.277 | 0 | -4.187 | -4.007 | -2.476 | 0.045 | -2.656 | -2.926 |
| | Signific | | ** | | ** | ** | * | | ** | ** |
| **Sugarcane** | $Z_c$ | 3.017 | 3.467 | -3.377 | -2.116 | -2.476 | -3.557 | -3.737 | -3.647 | -3.557 |
| | Signific | ** | ** | ** | * | * | ** | ** | ** | ** |
| **Sugar** | $Z_c$ | 3.647 | 4.727 | -0.045 | -4.457 | -4.457 | -4.457 | | | -4.457 |
| **beet** | Signific | ** | ** | | ** | ** | ** | | | ** |
| **Apple** | $Z_c$ | 3.197 | 5.358 | -4.907 | -5.088 | -4.997 | -3.557 | -2.926 | -3.557 | -3.557 |
| | Signific | ** | ** | ** | ** | ** | ** | ** | ** | ** |
| **Citrus** | $Z_c$ | 1.576 | 5.178 | -4.997 | -4.817 | -4.997 | -3.737 | -3.107 | -3.647 | -3.647 |
| | Signific | | ** | ** | ** | ** | ** | ** | ** | ** |
| **Tobacco** | $Z_c$ | 4.817 | 2.926 | -0.855 | -2.836 | -2.746 | -4.817 | -3.917 | -4.997 | -4.817 |
| | Signific | ** | ** | | ** | ** | ** | ** | ** | ** |

 **\* Significant at p < 0.05, \*\* significant at p < 0.01**


**Table 4. Moran's I test for production-based water footprint (PWF) and economic value-based water footprint (EWF) of crop**
**production.**

|  |  |  | Moran's *I* | Z-score | *p*-value |
|---|---|---|---|---|---|
|  |  | **2001** | 0.559 | 5.141 | 0.001 |
|  |  | **2006** | 0.227 | 2.207 | 0.014 |
|  | **Grain crop** | **2011** | 0.126 | 1.491 | 0.077 |
|  |  | **2016** | 0.214 | 2.085 | 0.021 |
| **PWF** |  | **2001-2016** | 0.263 | 2.659 | 0.009 |
| **(m³ kg⁻¹)** |  | **2001** | 0.302 | 2.972 | 0.004 |
|  |  | **2006** | 0.152 | 1.665 | 0.052 |
|  | **Cash crop** | **2011** | 0.094 | 1.252 | 0.106 |
|  |  | **2016** | 0.11 | 1.224 | 0.11 |
|  |  | **2001-2016** | 0.163 | 1.756 | 0.05 |
|  |  | **2001** | 0.585 | 5.392 | 0.001 |
|  |  | **2006** | 0.395 | 3.887 | 0.001 |
|  | **Grain crop** | **2011** | 0.311 | 3.073 | 0.003 |
|  |  | **2016** | 0.618 | 5.393 | 0.001 |
| **EWF** |  | **2001-2016** | 0.482 | 4.518 | 0.001 |
| **(m³ USD⁻¹)** |  | **2001** | -0.009 | 0.184 | 0.411 |
|  |  | **2006** | 0.04 | 0.653 | 0.24 |
|  | **Cash crop** | **2011** | 0.139 | 1.501 | 0.066 |
|  |  | **2016** | -0.145 | -0.914 | 0.187 |
|  |  | **2001-2016** | 0.016 | 0.418 | 0.307 |



**Table 5. Comparison between production-based water footprint (PWF) of crops production in mainland China in the current study and previous studies.**

| | PWF$_b$ (m$^3$ kg$^{-1}$) | | | PWF$_g$ (m$^3$ kg$^{-1}$) | | | PWF (m$^3$ kg$^{-1}$) | | |
|---|---|---|---|---|---|---|---|---|---|
| | Current | Mekonnen | Zhuo et al. | Current | Mekonnen | Zhuo et al. | Current | Mekonnen | Zhuo et al. |
| | 2001-2016 | 1996-2005 | 2008 | 2001-2016 | 1996-2005 | 2008 | 2001-2016 | 1996-2005 | 2008 |
| **Winter wheat** | 0.49 | 0.47 | 0.31 | 0.73 | 0.82 | 0.84 | 1.22 | 1.29 | 1.15 |
| **Spring wheat** | 1.03 | | | 0.56 | | | 1.59 | | |
| **Spring maize** | 0.27 | 0.07 | 0.07 | 0.65 | 0.79 | 0.75 | 0.92 | 0.86 | 0.82 |
| **Summer maize** | 0.25 | | | 0.73 | | | 0.98 | | |
| **Rice** | 0.36 | 0.25 | 0.38 | 0.46 | 0.55 | 0.96 | 0.82 | 0.80 | 1.34 |
| **Soybean** | 0.53 | 0.25 | 0.11 | 2.34 | 2.55 | 2.02 | 2.87 | 2.80 | 2.13 |
| **Groundnut** | 0.38 | 0.09 | 0.19 | 1.21 | 1.38 | 1.35 | 1.59 | 1.47 | 1.54 |
| **Rapeseed** | 0.00 | 0.00 | 0.00 | 1.18 | 1.39 | 1.74 | 1.18 | 1.39 | 1.74 |
| **Cotton** | 1.06 | 0.56 | | 3.58 | 3.26 | | 4.64 | 3.82 | |
| **Sugarcane** | 0.01 | 0.01 | 0.00 | 0.10 | 0.17 | 0.12 | 0.11 | 0.18 | 0.12 |
| **Sugar beet** | 0.00 | 0.00 | 0.00 | 0.10 | 0.15 | 0.07 | 0.10 | 0.15 | 0.07 |
| **Apple** | 0.04 | 0.03 | 0.04 | 0.35 | 0.80 | 0.31 | 0.39 | 0.83 | 0.35 |
| **Citrus** | 0.09 | 0.02 | | 0.63 | 0.45 | | 0.72 | 0.47 | |
| **Tobacco** | 0.23 | 0.25 | 0.01 | 1.67 | 2.01 | 1.63 | 1.90 | 2.26 | 1.64 |


**Table 6. Comparison between economic value-based water footprint (EWF) in the current results and previous studies.**

| Reference | Case | Year/Period | EWF$_{b,ir}$ m$^3$ USD$^{-1}$ | EWF$_{g,ir}$ m$^3$ USD$^{-1}$ | EWF$_{g,rf}$ m$^3$ USD$^{-1}$ | EWF m$^3$ USD$^{-1}$ |
|---|---|---|---|---|---|---|
| Schyns and Hoekstra (2014) | Wheat in Morocco | 1996-2005 | | | | 12.50 |
| Chouchane et al. (2015) | Wheat in Tunisia | 1996-2005 | 8.33 | 11.11 | 10.00 | 10.00 |
| Current study | Winter wheat in China | 2001-2016 | 17.57 | 3.82 | 6.63 | 6.81 |
| | Spring wheat in China | | 18.93 | 3.86 | 7.87 | 8.81 |
