# Peer review of "Physical versus economic water footprints in crop production: a 2 spatial and temporal analysis for China"

_Hydrology and Earth System Sciences, 2020_

## Referee Comment (RC1) · Anonymous Referee #1 · 17 May 2020

**Comments on "Physical versus economic water footprints in crop production: a case study for China" by Yang et al.**

The study assesses the WF of 14 crops from 2001 to 2016 in both physical and economic terms for 31 provinces of China. It also analyzes the spatial agglomeration and the temporal trends of the WFs. This provides valuable information as to which crops have a higher economic return per unit of water consumed. However, it is not clear why some of the analyzes are needed and there is a lack of in-depth discussion on the interpretation of the outcome. I suggest that the authors address the following comments before the paper got accepted:

**1 Subsection 2.2: The authors modeled crop yield using the AquaCrop model but used statistical output. The question is, why didn't you use the modeled yield? Don't you trust the outcome of your modeling? It is understandable that yield modeling has large uncertainty and may require a rigorous calibration of the model. Modeled ET and yield are consistent as both derive from the model, but to what degree is the ET consistent with the statistical yield? How reliable is the statistical yield?**

On line 88, I see that the modeled yield was checked against the provincial statistical yield. How good was the modeled yield compared to the statistical yield? I suggest that you plot the modeled vs. statistical yield to show the fitting between the two results. You can add the graph for each crop as additional information. Please explain if you did some manipulation on the modeled yield so it matches the statistical yield.

**2. Line 179 – 183: The interpretation of the SI is confusing. It seems to suggest larger SI to be better as province with larger SI are deemed to have "… (less water consumption per yield and higher economic benefits per water consumption unit). The SI is derived by comparing provincial WF and national average values. The result would thus mean, the province performs better in terms of generating higher economic benefits per unit of water. I would expect the SI value would be different if you compare two high performing provinces or evaluate the SI against a benchmark value instead of the national average value.**

**3. Please provide for each crop the yield, CWU, PWF, EWF for the irrigated and rainfed systems separately. This will help to see if there is difference in the economic WP of rainfed and irrigated systems.**

**4. The purpose of some the analyses are not clear e.g. Mann-Kendall. For the current study we don't need this analysis as the positive trend is visually clear from the figures. The authors themselves have used the test result on only one sentence (line 218-219).**

**5. The discussion is more on comparing the WF of the provinces and saying this WF is larger here and there (Line 226-247 and 371-400). The reader can read this fact from the figures. Please expand the discussion of the result and explain why the WF is large in one province and small in another. Is it climate, crop varieties, or what? Substantiate your argument with some references.**

**6. Please explain why there is spatial agglomeration of the EWP for the grain crops while none for the cash crops? What are the reasons for spatial agglomeration and what does it explain? Generally, provinces in the same climate**

region will have more or less similar WF. The price of the crops may also be dependent on the total production volume, demand for the crop, availability of market. Or are there other factors that play? Please discuss.

**7. Line 442-443: the statement seems to suggest that to improve the green water, rain water harvesting and storage should be improved. Is rainwater harvested green water or blue water? You need to be clear what you mean by the rainwater harvesting. If the farmer builds small retention pond to collect rainwater, the farmer is collecting blue water not green. But if a farmer manages his field to increase the water retention through tillage system and mulching, this is increasing the green water.  Please clarify your suggestions.**

**8. Line 444-445: the statement "As for northern China, green water (rain water) should be converted into blue water (irrigation water) as far as possible, so as to reduce blue water consumption while ensuring and increasing economic benefits." is not clear. What do you mean the green should be converted to blue? How do you convert green to blue? Do you mean, we need to increase irrigation?**

**9. Line 446-448: the statement is an empty statement: " The necessary way to alleviate the contradiction between water resource consumption and economic value creation is to adjust the agricultural production mode and the irrigation method according to local conditions." What do you find from your study and what practical ways do you suggest? How do farmers or policy makers adjust the agricultural production mode?**

**10. There is a statement in a number of places (lines 186, 371, 447, 468, 492) that reads, "contradiction between water consumption and economic value creation". There is no contradiction between water consumption and value creation. You cannot create value without water consumption. The issue should be how we optimize the value creation per unit of consumed water. Please rephrase your sentences.**

**11. On Table 4, the $EWP_{g,ir}$ is almost half of the $EWP_{g,rf}$. Why is that? Is the equivalent rainfed yield under the irrigated condition double that of the rainfed yield? Or the $CWU_{g,ir}$ is half of the $CWU_{g,rf}$? Generally, $CWU_{g,ir}$ is slightly lower than $CWU_{g,rf}$ but cannot be close to half. Please explain.**

Minor comments

**Please provide the spatial scale of the analysis in the last paragraph of the introduction section. I see in the discussion section that the analysis was done at a meteorological station level. How many stations per province?**

**Please provide the definition of "synergy evaluation index", what it does and how to interpret the result.**

**Line 30-31: change occupation to consumption on the following sentence**
 "The water footprint (WF) (Hoekstra, 2003) reveals the occupation and pollution of water in the process of production or consumption and …."

**Lin 52: the sentence is not clear - what do you mean by "WF coordination"?**
**Lin 57: Remove the period before (Tilman et al., 2011;Gao and Bryan, 2017;Cui et al., 2018)**

**Line 100: add reference to Hoekstra (2019)**

References

Hoekstra AY (2019) Green-blue water accounting in a soil water balance. AdWR 129:112-117. doi:https://doi.org/10.1016/j.advwatres.2019.05.012

---

## Referee Comment (RC2) · Anonymous Referee #2 · 19 May 2020

Comments on Physical versus economic water footprints in crop production a case study for China by Yang et al.

Summary

The manuscript evaluates the physical and economic Water Footprint of 14 crops categorised into cash and grain crops, between 2001 to 2016 over 31 provinces in China. A good background of existing studies is covered and it is shown what the added value of this study can be. However, I believe that there are some aspects that are clearly described or discussed. The methodology and discussion of the results needs to be clearly structured and expanded prior to acceptance of the paper.

General comments

1. In the methodology, split the models used in the analysis from the equations and calculation used to for PWF and EWF. Have a separate section prior to the calculations for the data and models used in the study and then move onto the calculations of the PWF and EWF which are a results of these results. This would include the AquaCrop model, the WF calculation frame and mode of soil water dynamic balance. Also would include a description of the national statistical data used. Otherwise, it is a little difficult to follow. You should also specifically show your equations for ETb and ETg.

2. You also state several times that the PWF and EWP together provide a measurement to analyse the synergy between water consumption of crop production and economic value creation, can you please explain how this happens in the introduction. And also explain why it is important.

3. You should also define what is blue and green water. Also why is green water rainfed and both blue and green water considered in irrigated systems. This might be clear to us but may not be clear to everyone who reads your article. This can either be included in the introduction or in the methodology section.

4. I do not really understand why you used the actual yield in your calculations instead of the modelled yield. You used the modelled yield for alpha but not for the actual yield which only effects your EWF for irrigated areas. Is there a reason for this? Also, maybe it would be a good idea to also calculate this using the modelled yield and compare with the results you obtain using the statistics or actual yield?

5. What is GeoDa? I have looked it up but I would suggest that you also describe this in your models section that I suggested you incorporate into your methodology.

6. More explanation is required in the results or in the discussion regarding what the results mean and not just the statement of the numbers. Why is the SI lower in one province, what significance does having H-H clustering in several provinces mean to the area? What does it mean if there is a decrease in agglomeration over time? What impact does this have? You need to go into the impacts of your numbers and trends so that the reader can get some information from the paper. These are just some questions but you should do this for all your results.

7. In table 8, you are comparing EWF in 'Wheat in Morocco', 'Wheat in Tunisia', 'Winter wheat in China' and 'Spring wheat in China'. Considering the large differences in the regions/countries, is this possible to compare? Please also make note these comparisons are not in the same regions and make sure you include the assumptions you make in these comparisons.

8. The main goal of this paper is the SI, but I am still unclear on this index. I think this index needs to be explained in greater detail in the methodology as well as the interpretation and

impact of this index in the subsequent results and discussion sections needs further improvement. This is the innovation of your paper so this needs to be more clear.

9. You need to be careful with some English terminology such as 'contradiction'. I do not believe that is what you meant in several places where you use it.

Specific comments

P3L79 – Change 'which are respectively calculated from the daily green ($ET_{g[t]}$, mm) and blue evapotranspiration ($ET_{b[t]}$, mm)' to which are respectively calculated from the blue evapotranspiration ($ET_{b[t]}$, mm) and daily green evapotranspiration ($ET_{g[t]}$, mm)' as you use respectively and then change the order or green and blue.

P5L131 – you should define what economic benefit unit is.

P7L184 – move the data section before your methods.

P7L175 – You state 'Obviously, -2 <= SI <= 2'. Why is this obvious? Please explain and clarify in text.

P8L191 – I would refer to your figure 1.

---

## Author Comment (AC1) · 8 Jun 2020

Authors' responses to Interactive comments on "Physical versus economic water footprints in crop production: a case study for China"

Dear Referee #1,

Thank you very much for your valuable comments and suggestions on our manuscript. We have provided our responses directly below each of the comments.

Anonymous Referee #1

The study assesses the WF of 14 crops from 2001 to 2016 in both physical and economic terms for 31 provinces of China. It also analyzes the spatial agglomeration and the temporal trends of the WFs. This provides valuable information as to which crops have a higher economic return per unit of water consumed. However, it is not clear why some of the analyzes are needed and there is a lack of in-depth discussion on the interpretation of the outcome. I suggest that the authors address the following comments before the paper got accepted:

**Response:** We deeply appreciate your positive words and valuable comments.

**1 Subsection 2.2: The authors modeled crop yield using the AquaCrop model but used statistical output. The question is, why didn't you use the modeled yield? Don't you trust the outcome of your modeling? It is understandable that yield modeling has large uncertainty and may require a rigorous calibration of the model. Modeled ET and yield are consistent as both derive from the model, but to what degree is the ET consistent with the statistical yield? How reliable is the statistical yield?**

On line 88, I see that the modeled yield was checked against the provincial statistical yield. How good was the modeled yield compared to the statistical yield? I suggest that you plot the modeled vs. statistical yield to show the fitting between the two results. You can add the graph for each crop as additional information. Please explain if you did some manipulation on the modeled yield so it matches the statistical yield.

**Response:** We are very sorry for our unclear expression and use of incorrect word.

The current calculation of water footprint is based on both the modeled ET and yield. Being consistent with the existing calibration method which has been widely applied (Mekonnen and Hoekstra, 2011; Zhuo et al., 2016a; Zhuo et al., 2016b; Wang et al., 2019; Zhuo et al., 2019), the modeled crop yield was calibrated at provincial level according to the statistics (NBSC, 2019). Within a province, we calibrated the average level of the modeled yields among station points to match the provincial statistics. Therefore, we kept the spatial variation in crop yields, so that in associated water footprints simulated by AquaCrop model. For sure, the calibrated yield was consistent with the modeled ET.

**2. Line 179 — 183: The interpretation of the SI is confusing. It seems to suggest larger SI to be better as province with larger SI are deemed to have "... (less water consumption per yield and higher economic benefits per water consumption unit). The SI is derived by comparing provincial WF and national average values. The result would thus mean, the province performs better in terms of generating higher economic benefits per unit of water. I would expect the SI value would be different if you compare two high performing provinces or evaluate the SI against a benchmark value instead of the national average value.**

**Response:** We strongly agree with you that a reasonable reference value should be used for synergy evaluation between crop PWF and EWF. The choice of reference value is based on the purpose of the evaluation. In the current study, the SI measures, considering the spatial heterogeneities in crop WFs among provinces, the synergy levels between the current PWF and EWF. The synergy (both the PWF and EWF are lower than the national averages), trade-off (one is higher than the national average while the other is lower), or lose-lose (both are higher than the national averages) situation can be identified. The most optimized situation means high economic value generated by low water consumption. If the reference value is set by the WF benchmark (Mekonnen and Hoekstra, 2014), then the SI will show information on efficiency. The meaning is totally different from the current one. But we believe it will be a nice study for the future that we like to carry out. We will add the discussion on the possible recommendations on further studies and wider applications of the proposed SI approach in the revised manuscript.

**3. Please provide for each crop the yield, CWU, PWF, EWF for the irrigated and rainfed systems separately. This will help to see if there is difference in the economic WP of rainfed and irrigated systems.**

**Response:** Yes, we will add a table consisting of the suggested information in the revised manuscript.

**4. The purpose of some the analyses are not clear e.g. Mann-Kendall. For the current study we don't need this analysis as the positive trend is visually clear from the figures. The authors themselves have used the test result on only one sentence (line 218-219).**

**Response:** Thank you very much for your valuable comments. We realize that the description of Mann-Kendall (M-K) test results is too little, so we will add the explanation of M-K in the relevant paragraphs.

For example, as for line 218-219: "The M-K test results of each crop's PWF in table 2 further confirm the above views.", we can visually see the downward trend of PWF and upward trend of yield for both grain and cash crops at the national average level from Figure 2. However, the grain crop here was the weighted average result of six different crops (winter wheat, spring wheat, spring maize, summer maize, rice and soybean) and the cash crop was the weighted average result of eight different crops (cotton, groundnut, rapeseed, sugar beet, sugarcane, citrus, apple, and tobacco). The main cultivated crops in different provinces were different, and the PWF and yield of different crops had different time evolution trends, and the time evolution trends of the $PWF_b$ and $PWF_g$ of the same crop were also different. Therefore,

we used M-K test for 14 crops to show more details. In Table 2, among grain crops, winter wheat had the lowest M-K statistical value in PWF (-4.547) showing an obvious downward trend, while the M-K statistic value of soybean was only -0.675, which meant that the PWF of soybean had almost no significant change.

We will improve the interpretation in relative places also including line 301-302: "The M-K test results in Table 7 further confirmed the above results, as the M-K statistical values of all crops' EWF passed the significance level test of p<0.05."

[Figure]

**Figure 2: Interannual variability of national average production-based water footprint (PWF) of (a) grain and (b) cash crops in China over 2001-2016.**

**Table 2. M-K analysis of production-based water footprint (PWF) of the 14 crops.**

| | | PWF (m³ kg⁻¹) | PWFb (m³ kg⁻¹) | PWFg (m³ kg⁻¹) | Yield (t ha⁻¹) |
|---|---|---|---|---|---|
| Winter Wheat | $Z_c$ | -4.547 | -3.737 | -4.547 | 5.178 |
| | Signific | ** | ** | ** | ** |
| Spring Wheat | $Z_c$ | -2.476 | -0.135 | -3.107 | 4.457 |
| | Signific | * | | ** | ** |
| Spring Maize | $Z_c$ | -4.097 | -4.097 | -2.476 | 3.647 |
| | Signific | ** | ** | * | ** |
| Summer Maize | $Z_c$ | -3.287 | -3.647 | -3.197 | 4.277 |
| | Signific | ** | ** | ** | ** |
| Rice | $Z_c$ | -3.377 | -3.107 | -3.017 | 4.637 |
| | Signific | ** | ** | ** | ** |
| Soybean | $Z_c$ | -0.675 | 1.846 | -1.396 | 1.126 |
| | Signific | | | | |
| Groundnut | $Z_c$ | -3.917 | -3.467 | -3.287 | 4.547 |
| | Signific | ** | ** | ** | ** |
| Rapeseed | $Z_c$ | -2.476 | 2.386 | -2.476 | 4.097 |
| | Signific | * | * | * | ** |
| Cotton | $Z_c$ | -4.007 | 0 | -4.187 | 4.277 |
| | Signific | ** | | ** | ** |
| Sugarcane | $Z_c$ | -2.476 | -3.377 | -2.116 | 3.467 |
| | Signific | * | ** | * | ** |
| Sugar beet | $Z_c$ | -4.457 | -0.045 | -4.457 | 4.727 |

| | | | | | |
|---|---|---|---|---|---|
| | Signific | ** | | ** | ** |
| Apple | $Z_c$ | -4.997 | -4.907 | -5.088 | 5.358 |
| | Signific | ** | ** | ** | ** |
| Citrus | $Z_c$ | -4.997 | -4.997 | -4.817 | 5.178 |
| | Signific | ** | ** | ** | ** |
| | $Z_c$ | -2.746 | -0.855 | -2.836 | 2.926 |
| Tobacco | Signific | ** | | ** | ** |

* Significant at $p < 0.05$, ** significant at $p < 0.01$

**5. The discussion is more on comparing the WF of the provinces and saying this WF is larger here and there (Line 226-247 and 371-400). The reader can read this fact from the figures. Please expand the discussion of the result and explain why the WF is large in one province and small in another. Is it climate, crop varieties, or what? Substantiate your argument with some references.**

**Response:** Thank you very much for your suggestions. We recognize that the current description on results does not go far enough. Also, as Referee #2 pointed out, we should give more explanation about results not just the statement of the numbers. We will identify the main reasons behind these results in terms of variations in possible impacting factors including climate, cultivation scales, agricultural productivities, or cultivation distribution via substantiation by existing references. We will revise and shorten the current lengthy and inefficient sentences and expand the discussion focusing on explanation how the results came from in the revised manuscript.

**6. Please explain why there is spatial agglomeration of the EWP for the grain crops while none for the cash crops?**

What are the reasons for spatial agglomeration and what does it explain? Generally, provinces in the same climate region will have more or less similar WF. The price of the crops may also be dependent on the total production volume, demand for the crop, availability of market. Or are there other factors that play? Please discuss.

**Response:** Yes, for a same crop, the spatial variations of its PWF are defined by climate and productivity. The price is one of the main factors defining the EWF. While in related to the cluster maps shown in the current results for grain and cash crops, the main factor is the cultivation distribution. Regarding the grain crops, the cultivation distributions of major grain crops in China show obvious spatial agglomeration characteristics. For instance, rice is mainly distributed in central and southern China (Hubei, Hunan, Jiangxi, Guangdong and Guangxi). Winter wheat is concentrated in Huang-Huai-Hai Plain (Shandong, Henan, Jiangsu, Anhui and Hebei). Whereas regarding the cash crops, the dominant crop differs among provinces which resulted in obvious scattered characteristics in related WFs. For example, in the northwest regions, there is only Xinjiang where cotton is planted on a large scale, and almost no cotton is planted in the surrounding provinces.

In addition, crop prices in the main producing provinces are generally lower, while vary affected by the regional economic level. For example, both Henan and Shandong are the main producing areas of winter wheat, but the price

(0.21 USD/kg in 2016) in Shandong, which has a more developed economy, was higher than that in Henan (0.17 USD/kg).

We will carefully analyze further the displayed results and add discussion on the reasons behind to improve the manuscript.

**7. Line 442-443: the statement seems to suggest that to improve the green water, rain water harvesting and storage should be improved. Is rainwater harvested green water or blue water? You need to be clear what you mean by the rainwater harvesting. If the farmer builds small retention pond to collect rainwater, the farmer is collecting blue water not green. But if a farmer manages his field to increase the water retention through tillage system and mulching, this is increasing the green water. Please clarify your suggestions.**

**Response:** Yes, we should clarify the green and blue water in statements. Sure, water supplied by rainwater harvesting is blue water (Hoekstra, 2019).Therefore, we will rewrite the pointed sentence like "Therefore, the utilisation efficiency of green water resources should be improved through water retention by tillage system and mulching. Meanwhile, more blue water can be generated through rainwater harvesting (Hoekstra, 2019). ".

**8. Line 444-445: the statement "As for northern China, green water (rain water) should be converted into blue water (irrigation water) as far as possible, so as to reduce blue water consumption while ensuring and increasing economic benefits." is not clear. What do you mean the green should be converted to blue? How do you convert green to blue? Do you mean, we need to increase irrigation?**

**Response:** We are sorry for the confusion because of unclear writing. Specifically, we mean two measures to increase the blue water efficiency in northern China. One is the rainwater harvesting in rainy season, especially for the short-time heavy rain which cannot effectively used by crops but easily cause soil erosion. It is the process of transferring green to blue water. The other one is reducing blue water consumption and loss at field by popularizing water-saving irrigation techniques and mulching practices. Such measure is helpful to improve the utilisation efficiency of both blue and green water.

We will clarify the relative discussion in the revised manuscript.

**9. Line 446-448: the statement is an empty statement: " The necessary way to alleviate the contradiction between water resource consumption and economic value creation is to adjust the agricultural production mode and the irrigation method according to local conditions." What do you find from your study and what practical ways do you suggest? How do farmers or policy makers adjust the agricultural production mode?**

**Response:** Thank you very much for your comments. Based on the current results, we recommend the government to improve agricultural water use efficiency through the application of water-saving irrigation techniques and better

management of all agricultural inputs, especially in northwest China. High water consumption and low economic value crops' acreages in non-primary production areas should be reduced. For the southern regions with abundant rainwater resources, the economic benefits of irrigation are very limited, on the contrary, rain-fed agriculture has obvious advantages and the potential to increase economic benefits. Therefore, farmers should improve the water conservation rate and the utilization efficiency of green water through farming system and coverage to reduce the amount of water used for irrigation. The government should also give financial subsidies for agricultural production to those provinces where there were lose-lose relationships between reducing the water resources input for harvesting crop yields and optimizing the economic benefits per unit of water consumption. Finally, it is necessary to improve the field managements especially in utilization rate of chemical fertilizers and pesticides to increase agricultural productivity further. Previous studies have shown that less than 50% of the fertilizer applied to fields actually fertilizes the crops for which it was intended, with the rest leaching into the environment (Zhang et al., 2013).

We will think more deeply and add more references to come up with more realistic proposals in the revised manuscript.

**10. There is a statement in a number of places (lines 186, 371, 447, 468, 492) that reads, "contradiction between water consumption and economic value creation". There is no contradiction between water consumption and value creation. You cannot create value without water consumption. The issue should be how we optimize the value creation per unit of consumed water. Please rephrase your sentences.**

**Response:** We are very sorry for the loose expression. It is also pointed by Referee#2. In the revision, we will check through the text and correct the word "contradiction" into "trade-off" or "lose-lose" accordingly.

**11. On Table 4, the $EWP_{g,ir}$ is almost half of the $EWP_{g,rf}$. Why is that? Is the equivalent rainfed yield under the irrigated condition double that of the rainfed yield? Or the $CWU_{g,ir}$ is half of the $CWU_{g,rf}$? Generally, $CWU_{g,ir}$ is slightly lower than $CWU_{g,rf}$ but cannot be close to half. Please explain.**

**Response:** Table R1 lists the average annual blue and green CWU and yield under irrigated and rain-fed condition by crops in China from 2001 to 2016. It can be seen that for all the crops, $CWU_{g,ir}$ was 23% (sugarcane) -51% (spring wheat) smaller than $CWU_{g,rf}$. Therefore, it is possible to result in $EWF_{g,ir}$ being much smaller than $EWF_{g,rf}$. We will add explanation on such results in the revised manuscript.

**Table R1. The average annual blue ($CWU_b$) and green crop water use ($CWU_g$) and yield (Y) at irrigated and rain-fed fields by crops in China from 2001 to 2016.**

| | Irrigated | | | Rain-fed | |
|---|---|---|---|---|---|
| | $CWU_{b,ir}$ (m³ ha$^{-1}$) | $CWU_{g,ir}$ (m³ ha$^{-1}$) | $Y_{IR}$ (t ha$^{-1}$) | $CWU_{g,rf}$ (m³ ha$^{-1}$) | $Y_{RF}$ (t ha$^{-1}$) |
| Winter Wheat | 3559 | 2694 | 5.13 | 4675 | 3.98 |
| Spring Wheat | 4448 | 1795 | 3.91 | 3661 | 2.60 |

| | | | | | |
|---|---|---|---|---|---|
| Spring Maize | 4022 | 2454 | 6.85 | 4504 | 5.22 |
| Summer Maize | 2543 | 2810 | 5.30 | 4605 | 4.99 |
| Rice | 2768 | 2705 | 6.64 | 4444 | 5.85 |
| Soybean | 3141 | 2721 | 2.10 | 4562 | 1.57 |
| Groundnut | 3466 | 2793 | 3.43 | 4662 | 3.23 |
| Rapeseed | | | | 2122 | 1.81 |
| Cotton | 4920 | 3007 | 1.50 | 5099 | 1.19 |
| Sugarcane | 4244 | 5534 | 123.71 | 7178 | 63.15 |
| Sugar beet | | | | 3818 | 39.64 |
| Apple | 4278 | 3439 | 16.68 | 5235 | 14.70 |
| Citrus | 3612 | 5073 | 12.39 | 7764 | 11.74 |
| Tobacco | 2204 | 2552 | 2.23 | 3622 | 1.99 |

Minor comments

**Please provide the spatial scale of the analysis in the last paragraph of the introduction section. I see in the discussion section that the analysis was done at a meteorological station level. How many stations per province?**

**Response:** Yes, we will add a table showing the number of meteorological stations per province in the revised manuscript.

**Please provide the definition of "synergy evaluation index", what it does and how to interpret the result.**

**Response:** We will add the following definition in the revised manuscript at the start of section 2.4.

The synergy evaluation index (SI) in the current study is the measure of the synergy levels between the PWF and EWP of crops, by summing up their corresponding difference between the water footprint and the base value divided by the range (the maximum minus the minimum) of the water footprint. Here, we adopt the national average level water footprint value as the reference for comparison.

**Line 30-31: change occupation to consumption on the following sentence**

"The water footprint (WF) (Hoekstra, 2003) reveals the occupation and pollution of water in the process of production or consumption and ...."

**Response:** We will correct the word in the revision.

**Lin 52: the sentence is not clear - what do you mean by "WF coordination"?**

**Response:** The WF coordination in current study indicates that the PWF and EWF of one province are both lower than the national averages, then it shows a good synergy in reducing the water resources input for harvesting crop yields and optimizing the economic benefits per unit of water consumption, compared with national average level.

**Lin 57: Remove the period before (Tilman et al., 201 I;Gao and Bryan, 2017;Cui et al., 2018)**

**Response:** We will do the correction in the revision.

**Line 100: add reference to Hoekstra (2019)**

**Response:** We will add the reference in proper places in the revision.

Reference

Hoekstra A Y (2019) Green-blue water accounting in a soil water balance. AdWR 129:112-117.
doi: https://doi.org/10.1016/j.advwatres.2019.05.012

**References**

Hoekstra, A. Y.: Green-blue water accounting in a soil water balance, Advances in Water Resources, 129, 112-117, https://doi.org/10.1016/j.advwatres.2019.05.012, 2019.

Mekonnen, M. M., and Hoekstra, A. Y.: The green, blue and grey water footprint of crops and derived crop products, Hydrology and Earth System Sciences, 15, 1577-1600, https://doi.org/10.5194/hess-15-1577-2011, 2011.

Mekonnen, M. M., and Hoekstra, A. Y.: Water footprint benchmarks for crop production: A first global assessment, Ecological Indicators, 46, 214-223, https://doi.org/10.1016/j.ecolind.2014.06.013, 2014.

NBSC: National Data. National Bureau of Statistics, Beijing, China, available at: http://data.stats.gov.cn/english/easyquery.htm?cn=E0103, 2019.

Wang, W., Zhuo, L., Li, M., Liu, Y., and Wu, P.: The effect of development in water-saving irrigation techniques on spatial-temporal variations in crop water footprint and benchmarking, Journal of Hydrology, 577, https://doi.org/10.1016/j.jhydrol.2019.123916, 2019.

Zhang, F., Chen, X., and Vitousek, P.: An experiment for the world, Nature, 497, 33-35, https://doi.org/10.1038/497033a, 2013.

Zhuo, L., Mekonnen, M. M., and Hoekstra, A. Y.: The effect of inter-annual variability of consumption, production, trade and climate on crop-related green and blue water footprints and inter-regional virtual water trade: A study for China (1978-2008), Water Res, 94, 73-85, https://doi.org/10.1016/j.watres.2016.02.037, 2016a.

Zhuo, L., Mekonnen, M. M., Hoekstra, A. Y., and Wada, Y.: Inter- and intra-annual variation of water footprint of crops and blue water scarcity in the Yellow River basin (1961–2009), Advances in Water Resources, 87, 29-41, https://doi.org/10.1016/j.advwatres.2015.11.002, 2016b.

Zhuo, L., Liu, Y., Yang, H., Hoekstra, A. Y., Liu, W., Cao, X., Wang, M., and Wu, P.: Water for maize for pigs for pork: An analysis of inter-provincial trade in China, Water Research, 166, https://doi.org/10.1016/j.watres.2019.115074, 2019.

---

## Author Comment (AC2) · 8 Jun 2020

Authors' responses to Interactive comments on "Physical versus economic water footprints in crop production: a case study for China"

Dear Referee #2,

We appreciate very much your valuable and helpful comments and suggestions concerning our manuscript. We have studied all the comments carefully and responded as followed.

Anonymous Referee #2

Summary

The manuscript evaluates the physical and economic Water Footprint of 14 crops categorised into cash and grain crops, between 2001 to 2016 over 31 provinces in China. A good background of existing studies is covered and it is shown what the added value of this study can be. However, I believe that there are some aspects that are clearly described or discussed. The methodology and discussion of the results needs to be clearly structured and expanded prior to acceptance of the paper.

**Response:** We are grateful for your positive comments and suggestions.

General comments

1. In the methodology, split the models used in the analysis from the equations and calculation used to for PWF and EWF. Have a separate section prior to the calculations for the data and models used in the study and then move onto the calculations of the PWF and EWF which are a results of these results. This would include the AquaCrop model, the WF calculation frame and mode of soil water dynamic balance. Also would include a description of the national statistical data used. Otherwise, it is a little difficult to follow. You should also specifically show your equations for ETb and ETg.

**Response:** We will add a section at the beginning of introduction of methodologies on the model used and data required, followed by the suggested logics in the revision.

The added section will be like:

**"2.1 AquaCrop modeling**

[revised manuscript text omitted]

The simulated yield per crop per year per station was calibrated at provincial level, by scaling the model outputs in order to fit provincial crop yield statistics (NBSC, 2019)."

2. You also state several times that the PWF and EWP together provide a measurement to analyse the synergy between water consumption of crop production and economic value creation, can you please explain how this happens in the introduction. And also explain why it is important.

**Response:** The economic benefits of water use form one important pillar of fresh water distribution (Hoekstra, 2014). However, traditional studies on agricultural efficient water use focus on crop water productivity from the physical perspective, and rarely make comprehensive evaluations combining the results with an economic perspective.

As the comprehensive index to evaluate types, quantities, and efficiency of water use in the process of crop production, the WF of crop production can be expressed based on either production (PWF, m$^3$ kg$^{-1}$) or economic value (EWF, m$^3$ per monetary unit) (Garrido et al., 2010; Hoekstra et al., 2011), which unifies the measurement of the physical and economic levels. PWF and EWF provide insightful measurements for reducing the water resources input for harvesting crop yields and optimizing the economic benefits per unit of water consumption, respectively. Therefore, based on the quantification of PWF and EWF, we constructed the synergy evaluation index (SI) of water footprint, so that the original intention of the study -- comprehensive assessment from the perspective of both physics and economics can be implemented.

We will use the above explanation to elaborate further on the motivation for a comprehensive evaluation using both PWF and EWF in the introduction in the revised manuscript.

3. You should also define what is blue and green water. Also why is green water rainfed and both blue and green water considered in irrigated systems. This might be clear to us but may not be clear to everyone who reads your article. This can either be included in the introduction or in the methodology section.

**Response:** Blue water is surface and ground water, whereas green water is defined as the water kept in the unsaturated soil layer and precipitation, which is eventually transferred into canopy evapotranspiration (Falkenmark and Rockström, 2006). The consumptive WF of crop production can be divided into blue and green WFs (Hoekstra et al., 2011). The blue WF refers to the consumption of surface water and groundwater. In agriculture, the blue WF measures irrigation water consumption. Green WF refers to the consumption of rainwater (Hoekstra et al., 2011). Therefore, only green water is consumed at rainfed field.

We will add corresponding information to the revised introduction.

4. I do not really understand why you used the actual yield in your calculations instead of the modelled yield. You used the modelled yield for alpha but not for the actual yield which only effects your EWF for irrigated areas. Is there a reason for this? Also, maybe it would be a good idea to also calculate this using the modelled yield and compare with the results you obtain using the statistics or actual yield?

**Response:** We are very sorry to raise both your and Referee #1's confusion because of our unclear expression.

The current calculation of water footprint is based on both the modeled ET and yield. Being consistent with the existing calibration method which has been widely applied (Mekonnen and Hoekstra, 2011; Zhuo et al., 2016b, 2016c, 2019; Wang et al., 2019), the modeled crop yield was calibrated at provincial level according to the statistics (NBSC, 2019). Within a province, we calibrated the average level of the modeled yields among station points to match the provincial statistics. Therefore, we kept the spatial variation in crop yields, so that in associated water footprints simulated by AquaCrop model.

We will add the above explanation to the revised manuscript.

5. What is GeoDa? I have looked it up but I would suggest that you also describe this in your models section that I suggested you incorporate into your methodology.

**Response:** Thank you very much for your valuable suggestions.

GeoDa is a free software program intended to serve as a user-friendly and graphical introduction to spatial analysis. It includes functionality ranging from simple mapping to exploratory data analysis, the visualization of global and local spatial autocorrelation, and spatial regression. A key feature of GeoDa is an interactive environment that combines maps with statistical graphics, using the technology of dynamically linked windows. In terms of the range of spatial statistical techniques included, GeoDa is most alike to the collection of functions developed in the open-source R environment. (Anselin et al., 2006).

We will add the above description of GeoDa to methodology in the revised manuscript.

6. More explanation is required in the results or in the discussion regarding what the results mean and not just the statement of the numbers. Why is the SI lower in one province, what significance does having H-H clustering in several provinces mean to the area? What does it mean if there is a decrease in agglomeration over time? What impact does this have? You need to go into the impacts of your numbers and trends so that the reader can get some information from the paper. These are just some questions but you should do this for all your results.

**Response:** Thank you very much for your valuable suggestions. We will carefully check all the results accordingly and improve the interpretation by deeper analysis on reasons behind the shown numbers.

For example, for the first question in the comment, we find that SI values negative mostly in northwest China for grain crops, whereas positive in southeast coastal provinces in China. The main reason behind is that the drier Northwest, where grows wheat and maize, have both high water intensity and low crop prices. While the water-abundant and economically developed southeast coastal provinces grow rice with a lower PWF and higher prices. In the revision, we will show in detail all the revised parts by addressing all the questions listed in the comments, in responses to comments.

7. In table 8, you are comparing EWF in 'Wheat in Morocco', 'Wheat in Tunisia', 'Winter wheat in China' and 'Spring wheat in China'. Considering the large differences in the regions/countries, is this possible to compare? Please also make note these comparisons are not in the same regions and make sure you include the assumptions you make in these comparisons.

**Response:** Yes, we realize that the current writing is not clear in related to Table 8. There was no existing EWF values for the same region. We wish to show the available values on EWF of crops, while for countries other than China. In the revised paper, we will clarify the statements in text and discuss the reasons behind the differences in EWF values among the countries.

8. The main goal of this paper is the SI, but I am still unclear on this index. I think this index needs to be explained in greater detail in the methodology as well as the interpretation and impact of this index in the subsequent results and discussion sections needs further improvement. This is the innovation of your paper so this needs to be more clear.

**Response:** As we respond to Referee #1's similar comments, the synergy evaluation index (SI) in the current study is the measure of the synergy levels between the PWF and EWF of crops, by summing up their corresponding difference between the water footprint and the base value divided by the range (the maximum minus the minimum) of the water footprint. Here, we adopt the weighted national average level water footprint value as the reference for comparison. The synergy (both the PWF and EWF are lower than the national averages), trade-off (one is higher than the national average while the other is lower), or lose-lose (both are higher than the national averages) situation can be identified.

For the two provinces with high SI values, they were both in an advantageous position, while the one with a higher SI values performed better in terms of synergy between PWF and EWF.

We will improve the interpretation on the SI in the methodology and clarify the impact of this index in the subsequent results and discussion sections in the revised manuscript.

9. You need to be careful with some English terminology such as 'contradiction'. I do not believe that is what you meant in several places where you use it.

**Response:** We are very sorry for the incorrect wording. As we respond to the similar comment by Referee #1, we will check carefully through the text and correct the word "contradiction" into "trade-off" or "lose-lose" accordingly.

Specific comments

P3L79 – Change 'which are respectively calculated from the daily green ($ET_{g[t]}$, mm) and blue evapotranspiration ($ET_{b[t]}$, mm)' to which are respectively calculated from the blue evapotranspiration ($ET_{b[t]}$, mm) and daily green evapotranspiration ($ET_{g[t]}$, mm)' as you use respectively and then change the order or green and blue.

**Response:** We will correct the sentence in the revision.

P5L131 – you should define what economic benefit unit is.

**Response:** The economic benefit unit refers to crop price in the current study. We will clarify the terminology in the revision.

P7L184 – move the data section before your methods.

**Response:** We will revise accordingly.

P7L175 – You state 'Obviously, -2 <= SI <= 2'. Why is this obvious? Please explain and clarify in text.

**Response:** The synergy evaluation index (SI) is the sum of the difference between the water footprint and the base value divided by the range (the maximum minus the minimum) of the water footprint. Here, we adopt the national average level water footprint value as the reference of comparison. The SI is calculated as follows:

$$SI_{i,j,c} = \frac{\overline{PWF_{j,c}} - PWF_{i,j,c}}{PWF_{j,c,max} - PWF_{j,c,min}} + \frac{\overline{EWF_{j,c}} - EWF_{i,j,c}}{EWF_{j,c,max} - EWF_{j,c,min}}$$ (18 in the original manuscript)

where $SI_{i,j,c}$ is the synergy evaluation index of PWF and EWF of crop c at province i in year j, $\overline{PWF_{j,c}}$(m³ kg⁻¹) and $\overline{EWF_{j,c}}$(m³ USD⁻¹) are the averages at the national level in year j. Obviously, the absolute value of the difference between the WF and their corresponding national average level cannot exceed the maximum minus minimum values. Therefore, the absolute value of SI cannot exceed 2.

We will clarify in the text in the revision.

P8L191 – I would refer to your figure 1.

**Response:** We will revise accordingly.

---

## Author Response (AR1)

**Physical versus economic water footprints in crop production: a case study for China**

Xi Yang, La Zhuo, Pengxuan Xie, Hongrong Huang, Bianbian Feng, Pute Wu

**Authors' responses to Referees' comments**

We greatly appreciate the opportunity to revise the study and the valuable comments and suggestions by two Referees. We have carefully learned and addressed all the comments. Please find below our detailed responses point by point. The revised parts are coloured in RED in the revised manuscript.

Referee #1

The study assesses the WF of 14 crops from 2001 to 2016 in both physical and economic terms for 31 provinces of China. It also analyzes the spatial agglomeration and the temporal trends of the WFs. This provides valuable information as to which crops have a higher economic return per unit of water consumed. However, it is not clear why some of the analyzes are needed and there is a lack of in-depth discussion on the interpretation of the outcome. I suggest that the authors address the following comments before the paper got accepted:

**Response:** We deeply appreciate your positive words and valuable comments.

**1 Subsection 2.2: The authors modeled crop yield using the AquaCrop model but used statistical output. The question is, why didn't you use the modeled yield? Don't you trust the outcome of your modeling? It is understandable that yield modeling has large uncertainty and may require a rigorous calibration of the model. Modeled ET and yield are consistent as both derive from the model, but to what degree is the ET consistent with the statistical yield? How reliable is the statistical yield?**

On line 88, I see that the modeled yield was checked against the provincial statistical yield. How good was the modeled yield compared to the statistical yield? I suggest that you plot the modeled vs. statistical yield to show the fitting between the two results. You can add the graph for each crop as additional information. Please explain if you did some manipulation on the modeled yield so it matches the statistical yield.

**Response:** We are very sorry for our unclear expression and use of incorrect word.

The current calculation of water footprint is based on both the modeled ET and yield. Being consistent with the existing calibration method which has been widely applied (Mekonnen and Hoekstra, 2011; Zhuo et al., 2016a; Zhuo et al., 2016c; Wang et al., 2019; Zhuo et al., 2019), the modeled crop yield was calibrated at provincial level according to the statistics (NBSC, 2019). Within a province, we calibrated the average level of the modeled yields among station points to match the provincial statistics. Therefore, we kept the spatial variation in crop yields, so that in associated water footprints simulated by AquaCrop model. For sure, the calibrated yield was consistent with the modeled ET.

We added the above explanation in the Section 2.2 of the revised manuscript (Line 138-142).

#2. Line 179 — 183: The interpretation of the SI is confusing. It seems to suggest larger SI to be better as province with larger SI are deemed to have ". . . (less water consumption per yield and higher economic benefits per water consumption unit). The SI is derived by comparing provincial WF and national average values. The result would thus mean, the province performs better in terms of generating higher economic benefits per unit of water. I would expect the SI value would be different if you compare two high performing provinces or evaluate the SI against a benchmark value instead of the national average value.

**Response:** We strongly agree with you that a reasonable reference value should be used for synergy evaluation between crop PWF and EWF. The choice of reference value is based on the purpose of the evaluation.

In the current study, the SI measures, considering the spatial heterogeneities in crop WFs among provinces, the synergy levels between the current PWF and EWF. The synergy (both the PWF and EWF are lower than the national averages), trade-off (one is higher than the national average while the other is lower), or lose-lose (both are higher than the national averages) situation can be identified. The most optimized situation means high economic value generated by low water consumption. For the two provinces with high SI values, they were both in an advantageous position, while the one with a higher SI values performed better in terms of synergy between PWF and EWF. If the reference value is set by the WF benchmark (Mekonnen and Hoekstra, 2014), then the SI will show information on water use efficiencies from physical and economic perspectives. The meaning is totally different from the current one. But we believe it will be a nice study for the future that we like to carry out.

We added the above sentences in the end of Discussion (Line 542-549).

#3. Please provide for each crop the yield, CWU, PWF, EWF for the irrigated and rainfed systems separately. This will help to see if there is difference in the economic WP of rainfed and irrigated systems.

**Response:** Yes, considering that some data will appear in different tables repeatedly, we integrated the contents of Table 1 (National average production-based water footprint (PWF) of crops in China for the years 2001 and 2016) and Table 4 (National average economic value-based water footprint (EWF) of crops in China for the years 2001 and 2016) in the original manuscript, and added the suggested information to Table 2 (National average production-based water footprint (PWF) and economic value-based water footprint (EWF) of crops in China for the years 2001 and 2016) in the revised manuscript.

It should be noted that, combined with the questions raised by Referee #2, we found ourselves technical errors in the calculation of EWF under irrigation and rainfed conditions. Compared to rainfed agriculture, the ratio (α) of crop yield increment under full irrigation has been recalculated through the simulated crop yield after calibrated at provincial level under the rainfed and irrigation modes. We found that the recalculation did not affect the conclusion. We corrected the data related to EWF.

[revised manuscript text omitted]

**Response:** Thank you very much for your valuable comments. We realize that the description of Mann-Kendall (M-K) test results is insufficient. We added the explanation of M-K in the relevant paragraphs.

In the Section 3.1 we added (Line 257-265) "The PWF and yield of different crops had different temporal evolutions. The temporal trends in the $PWF_b$ and $PWF_g$ of a same crop were also different. Among grain crops, winter wheat had the lowest M-K statistical value in PWF (-4.547) and the highest in yield (5.178) jointly showing an obvious positive trend on improving water use efficiency. While the M-K statistic value of soybean was only -0.675, which meant that the PWF of soybean had little decrease. Soybean planting was dominated by individual farmer mode, with small and fragmented scales and a low planting mechanization degree. Moreover, the harvested area was shrunk (7,202 thousand hectares in 2016, 24% less than 2001). For cash crops, the changes of PWF and yield were most pronounced for fruit crops (apple and citrus). The M-K test result of $PWF_b$ of cotton with highest water consumption intensity was zero, with almost no changes, given little changes in the yield level at most cotton growing areas."

In the Section 3.2 we added (Line 360-367) "M-K test results for the EWF of most crops were at the similar significance level as for the corresponding PWF. It is mainly because the M-K test results of the prices of most crops were at the same significant level as the corresponding M-K test results of the yields. Due to the significantly increased price, the EWF M-K test result of soybean was -2.116, which was higher than the test result of corresponding PWF (-0.675). Cotton is another crop worthy of attention. M-K test result for EWF of cotton was -2.476, whose significance level was lower than that of PWF. This is mainly due to fluctuations in the price of cotton. In addition, it can be seen that the changes of $EWF_{b,ir}$ of most crops were not as obvious as those of $EWF_{g,ir}$ and $EWF_{g,ir}$. It indicates that there is more potential in optimizing the economic benefit of agricultural blue water input."

#5. The discussion is more on comparing the WF of the provinces and saying this WF is larger here and there (Line 226-247 and 371-400). The reader can read this fact from the figures. Please expand the discussion of the result and explain why the WF is large in one province and small in another. Is it climate, crop varieties, or what? Substantiate your argument with some references.

**Response:** Thank you very much for your suggestions. We recognize that the current description on results does not go far enough. Also, as Referee #2 pointed out, we should give more explanation about results instead of just the statement on the numbers. Following sentences on the reasons behind the shown results are added in the revised manuscript.

In the Section 3.1 Temporal and spatial evolution of PWF (Line 276-278) we added "The main reason behind is that the drier northwest, where grows wheat and maize, has relatively higher evapotranspiration so that higher PWF. While the water-abundant and wet southeast coastal provinces grow rice with a lower PWF."

For the phenomenon that "Differently from grain crop, the PWF of cash crop was higher in the Beijing-Tianjin-Hebei region, and lower in Inner Mongolia province and the southern areas (Guangdong, Guangxi and Hainan), without an obvious clustered characteristic", we added the reason (Line 286-290) that "This can be interpreted that regarding the cash crops, the dominant crop differs among provinces which resulted in obvious scattered characteristics in related WFs. For instance, cotton and groundnut with PWF of 3.68 $m^3$ $kg^{-1}$ and 1.49 $m^3$ $kg^{-1}$ (in 2016) were the leading cash crops in Beijing-Tianjin-Hebei region whereas rapeseed of lower PWF (1.04 $m^3$ $kg^{-1}$ in 2016) was the main cash crop in Inner Mongolia."

In the Section 3.2 Temporal and spatial evolution of EWF, for the phenomenon that "Generally, the EWF of grain crop was higher in Inner Mongolia and north-western China (Shaanxi, Gansu, Ningxia and Xinjiang); Guangdong, Jiangxi, Fujian, Zhejiang and other south-eastern coastal provinces were at a relatively low level", we added the explanation (Line 370-372) that "The northwest, with higher PWF, has lower crop prices due to the relatively underdeveloped economies. In contrast, the economically advanced southeast coastal provinces have both low crop water consumption and higher prices."

Regarding the pointed paragraph in Section 3.3 (Line 443-463). We found the misleading in the text on explanation of reasons because of lengthy sentences on showing only the high or low. We deleted these sentences for easier access to the words on explanation. We show here the revised paragraphs and the text on reason analysis are underlined.

"Taking 2016 as an example, we further look at the reasons for the "lose-lose" relationship between reducing the water resources input for harvesting crop yields and optimizing the economic benefits per unit of water consumption in both grain and cash crops (see Fig. 9), from the perspective of planting structure (see Fig. 10). Shaanxi province had the highest PWF in China (1.23 $m^3$ $kg^{-1}$), and the second highest EWF (7.48 $m^3$ $USD^{-1}$). In Shaanxi, winter wheat and spring maize with high water consumption and low yield accounted for more than 90% of the total sown area of grain crops, with yields lower than the national averages by 24% and 26%, respectively. Moreover, the price of wheat in Shaanxi province (0.17 USD $kg^{-1}$) was lower than the national average (0.19 USD $kg^{-1}$). The reasons for high water consumption per unit of grain production coupled with poor economic benefits in Shaanxi province can be attributed to the above two points. In contrast, in Jiangxi province, where rice, which has low water consumption intensity, is the main grain crop (rice accounting for 95% of the grain crops), PWF and EWF were 0.77 $m^3$ $kg^{-1}$ and 3.63 $m^3$ $USD^{-1}$, well below the national averages (0.93 $m^3$ $kg^{-1}$, 5.04 $m^3$ $USD^{-1}$).

As for cash crop, the PWF of Tianjin was 1.92 $m^3$ $kg^{-1}$, the highest in China, and the EWF was 3.26 $m^3$ $USD^{-1}$, the fifth highest in China, which was significantly higher than the national average (2.05 $m^3$ $USD^{-1}$). It can be seen from Fig. 10b that cotton accounted for the largest proportion (70%) in the planting structure of cash crops in Tianjin. Cotton consumed the most water per yield unit of cash crops, while the price unit of cotton in Tianjin was the second lowest in China (1.11 USD $kg^{-1}$), which did not reflect the advantage of cotton as a high-value crop. Jiangxi province showed the highest EWF in China (3.86 $m^3$ $USD^{-1}$), and a PWF (0.96 $m^3$ $kg^{-1}$) which was also higher than the national average (0.46 $m^3$ $kg^{-1}$). Figure 10b shows that citrus (planting area accounting for 29% of cash crops) and rapeseed (planting

area accounting for 48% of cash crops) are the main cash crops in Jiangxi. However, the price unit of citrus in Jiangxi was the third lowest (0.17 USD kg$^{-1}$, only 62% of the national average), and the yield of rapeseed was also the third lowest (1.34 t ha$^{-1}$, 32% lower than the national average). In contrast, the main cash crop in Shanxi was apple (planting area accounting for 87% of cash crops), with low water consumption intensity and a yield which was the second highest in China (28.5 t ha$^{-1}$), 1.5 times larger than the national average (18.9 t ha$^{-1}$)."

**6. Please explain why there is spatial agglomeration of the EWP for the grain crops while none for the cash crops? What are the reasons for spatial agglomeration and what does it explain? Generally, provinces in the same climate region will have more or less similar WF. The price of the crops may also be dependent on the total production volume, demand for the crop, availability of market. Or are there other factors that play? Please discuss.**

**Response:** For a same crop, the spatial variations of its PWF are defined by climate and productivity. The price is one of the main factors defining the EWF. While in related to the cluster maps shown in the current results for grain and cash crops, the main factor is the cultivation distribution. Regarding the grain crops, the cultivation distributions of major grain crops in China show obvious spatial agglomeration characteristics. For instance, rice is mainly distributed in central and southern China (Hubei, Hunan, Jiangxi, Guangdong and Guangxi). Winter wheat is concentrated in Huang-Huai-Hai Plain (Shandong, Henan, Jiangsu, Anhui and Hebei). Whereas regarding the cash crops, the dominant crop differs among provinces (see Fig. 10b) which resulted in obvious scattered characteristics in related WFs. For example, in the northwest regions, there is only Xinjiang where cotton is planted on a large scale, and almost no cotton is planted in the surrounding provinces.

In addition, crop prices in the main producing provinces are generally lower, while vary affected by the regional economic level. For example, both Henan and Shandong are the main producing areas of winter wheat, but the price (0.21 USD kg$^{-1}$ in 2016) in Shandong, which has a more developed economy, was higher than that in Henan (0.17 USD kg$^{-1}$).

We add the above discussion in the end Section 3.2 (Line 404-414).

**7. Line 442-443: the statement seems to suggest that to improve the green water, rain water harvesting and storage should be improved. Is rainwater harvested green water or blue water? You need to be clear what you mean by the rainwater harvesting. If the farmer builds small retention pond to collect rainwater, the farmer is collecting blue water not green. But if a farmer manages his field to increase the water retention through tillage system and mulching, this is increasing the green water. Please clarify your suggestions.**

**Response:** Yes, we should clarify the green and blue water in statements. Sure, water supplied by rainwater harvesting is blue water (Hoekstra, 2019). We rewrote the pointed sentence (Line 505-507) as "Therefore, the utilisation efficiency of green water resources should be improved through water retention by tillage system and mulching. Meanwhile, more blue water can be generated through rainwater harvesting (Hoekstra, 2019)."

#8. Line 444-445: the statement "As for northern China, green water (rain water) should be converted into blue water (irrigation water) as far as possible, so as to reduce blue water consumption while ensuring and increasing economic benefits." is not clear. What do you mean the green should be converted to blue? How do you convert green to blue? Do you mean, we need to increase irrigation?

**Response:** We are sorry for the confusion because of unclear writing. We rewrote the pointed sentences (Line 507-511) as "Specifically, we suggest two measures to increase the blue water efficiency in northern China. One is the rainwater harvesting in rainy season, especially for the short-time heavy rain which cannot effectively used by crops but easily cause soil erosion. The other one is reducing blue water consumption and loss at field by popularizing water-saving irrigation techniques and mulching practices. Such measure is helpful to improve the utilisation efficiency of both blue and green water."

#9. Line 446-448: the statement is an empty statement: " The necessary way to alleviate the contradiction between water resource consumption and economic value creation is to adjust the agricultural production mode and the irrigation method according to local conditions." What do you find from your study and what practical ways do you suggest? How do farmers or policy makers adjust the agricultural production mode?

**Response:** Thank you very much for your comments.

In the discussion part, we added (Line 512-521) "Based on the current results, we recommend the government to improve agricultural water use efficiency through the extension of water-saving irrigation techniques and better agricultural inputs management, especially in northwest China. High water consumption and low economic value crops' acreages in non-primary production areas should be reduced. For the southern regions with abundant rainwater resources, the economic benefits of irrigation are very limited, on the contrary, rainfed agriculture has obvious advantages and the potential to increase economic benefits. Therefore, farmers should improve the water conservation rate and the utilization efficiency of green water through farming system and coverage to reduce the amount of water used for irrigation. The government should also give financial subsidies for agricultural production to those provinces where there were lose-lose relationships between reducing the water resources input for harvesting crop yields and optimizing the economic benefits per unit of water consumption. Finally, improve the field managements especially in utilization rate of chemical fertilizers and pesticides to increase agricultural productivity further (Zhang et al., 2013)."

#10. There is a statement in a number of places (lines 186, 371, 447, 468, 492) that reads, "contradiction between water consumption and economic value creation". There is no contradiction between water consumption and value creation. You cannot create value without water consumption. The issue should be how we optimize the value creation per unit of consumed water. Please rephrase your sentences.

**Response:** We are very sorry for the loose expression. It is also pointed by Referee#2. In the revision, we checked through the text and corrected the word "contradiction" into "trade-off" or "lose-lose" accordingly.

**11. On Table 4, the $EWP_{g,ir}$ is almost half of the $EWP_{g,rf}$. Why is that? Is the equivalent rainfed yield under the irrigated condition double that of the rainfed yield? Or the $CWU_{g,ir}$ is half of the $CWU_{g,rf}$? Generally, $CWU_{g,ir}$ is slightly lower than $CWU_{g,rf}$ but cannot be close to half. Please explain.**

**Response:** In the revised manuscript, Table 2 lists the blue and green CWU and yield under irrigated and rainfed conditions by crops in China for the years 2001 and 2016. We added the following explanation:

In the Section 3.2 (Line 354-357): "Table 2 also lists the annual blue and green CWU and yield under irrigated and rainfed conditions by crops in China for the years 2001 and 2016. It can be seen that for all the crops, $CWU_{g,ir}$ was 21% (sugarcane) -55% (spring wheat) smaller than $CWU_{g,rf}$ in 2016. Therefore, it is possible to result in $EWF_{g,ir}$ being much smaller than $EWF_{g,rf}$."

Minor comments

**Please provide the spatial scale of the analysis in the last paragraph of the introduction section. I see in the discussion section that the analysis was done at a meteorological station level. How many stations per province?**

**Response:** Yes, we added Table 1 showing the number of meteorological stations per province in the revised manuscript. In the last paragraph of the Introduction (Line 77-80), we rewrote the sentence as "First, the blue and green PWF ($PWF_b$, $PWF_g$) of 14 major crops (winter wheat, spring wheat, spring maize, summer maize, rice, soybean, cotton, groundnut, rapeseed, sugar beet, sugarcane, citrus, apple, and tobacco) is calculated annually in 31 provinces at the meteorological station level, and the corresponding EWF is derived. Table 1 shows the number of meteorological stations per province."

**Table 1. Number of meteorological stations per province.**

| | Province | Number of weather stations | Climatic zone |
|---|---|---|---|
| North-central | Beijing | 3 | Temperate |
| | Tianjin | 3 | |
| | Shanxi | 27 | |
| Northeast | Inner Mongolia | 36 | Continental temperate & temperate |
| | Liaoning | 25 | |
| | Jilin | 29 | |
| | Heilongjiang | 34 | |
| Huang-Huai-Hai | Hebei | 19 | Temperate |
| | Shandong | 21 | |
| | Henan | 17 | |
| | Anhui | 21 | |
| Southeast | Shanghai | 1 | Sub-tropics |
| | Zhejiang | 21 | |
| | Fujian | 22 | |
| Yangtze (middle & lower reaches) | Jiangsu | 22 | Sub-tropics |
| | Jiangxi | 26 | |
| | Hubei | 27 | |
| | Hunan | 29 | |
| South-central | Guangdong | 36 | Sub-tropics & tropics |
| | Guangxi | 18 | |
| | Hainan | 5 | |
| Southwest | Chongqing | 11 | Sub-tropics |
| | Sichuan | 38 | |
| | Guizhou | 31 | |
| | Yunnan | 25 | |
| | Tibet | 17 | |

| Northwest | Shaanxi | 32 | Continental temperate & plateau and mountain |
| | Gansu | 23 | |
| | Qinghai | 25 | |
| | Ningxia | 12 | |
| | Xinjiang | 42 | |

**Please provide the definition of "synergy evaluation index", what it does and how to interpret the result.**

**Response:** We added the following definition in the revised manuscript at the start of Section 2.5 The synergy evaluation index (SI) of PWF and EWF.

The synergy evaluation index (SI) in the current study is the measure of the synergy levels between the PWF and EWF of crops, by summing up their corresponding difference between the water footprint and the base value divided by the range (the maximum minus the minimum) of the water footprint. Here, we adopt the national average level water footprint value as the reference for comparison (Line 211-214).

**Line 30-31: change occupation to consumption on the following sentence**

"The water footprint (WF) (Hoekstra, 2003) reveals the occupation and pollution of water in the process of production or consumption and ...."

**Response:** We corrected the word in the revision (Line 31).

**Lin 52: the sentence is not clear - what do you mean by "WF coordination"?**

**Response:** The WF coordination in current study indicates that the PWF and EWF of one province are both lower than the national averages, then it shows a good synergy in reducing the water resources input for harvesting crop yields and optimizing the economic benefits per unit of water consumption, compared with national average level. We add the explanation in the text (Line 57-59).

**Lin 57: Remove the period before (Tilman et al., 201 I;Gao and Bryan, 2017;Cui et al., 2018)**

 **Response:** We made the correction in the revision.

**Line 100: add reference to Hoekstra (2019)**

**Response:** We added the reference in proper places in the revision.

**Response:** Thank you very much for your valuable suggestions. As also commented by Referee #1, we carefully checked all the results accordingly and improved the interpretation by deeper analysis on reasons behind the shown numbers.

In the end Section 3.2 (Line 404-414), we added "For a same crop, the spatial variations of its PWF are defined by climate and productivity. The price is one of the main factors defining the EWF. While in related to the cluster maps shown in the current results for grain and cash crops, the main factor is the cultivation distribution. Regarding the grain crops, the cultivation distributions of major grain crops in China show obvious spatial agglomeration characteristics. For instance, rice is mainly distributed in central and southern China (Hubei, Hunan, Jiangxi, Guangdong and Guangxi). Winter wheat is concentrated in Huang-Huai-Hai Plain (Shandong, Henan, Jiangsu, Anhui and Hebei). Whereas regarding the cash crops, the dominant crop differs among provinces (see Fig. 10b) which resulted in obvious scattered characteristics in related WFs. For example, in the northwest regions, there is only Xinjiang where cotton is planted on a large scale, and almost no cotton is planted in the surrounding provinces. In addition, crop prices in the main producing provinces are generally lower, while vary affected by the regional economic level. For example, both Henan and Shandong are the main producing areas of winter wheat, but the price (0.21 USD kg$^{-1}$ in 2016) in Shandong, which has a more developed economy, was higher than that in Henan (0.17 USD kg$^{-1}$)."

Regarding the spatial differences in SI, we added the analysis in the Section 3.3 (Line 424-430) that "Overall, the SI of grain crop was negative in Inner Mongolia and north-western China (Shaanxi, Gansu, Ningxia), whereas in Guangdong, Jiangxi, Fujian, Zhejiang and other coastal areas in south-eastern China it was positive, with a clustered distribution. With the development of water-saving technologies and the improvement of agricultural management, China has made gratifying progress in the efficient use of water for crop production from a single physical or economic perspective. However, only by combining the physical and economic perspectives can we gain a deeper understanding of the underlying problems and catch the synergies, trade-offs and even lose-lose relationships between reducing the water resources input for harvesting crop yields and optimizing the economic benefits per unit of water consumption in different regions."

In Section Discussion, we improved the statements on implementations of the current results in terms of reducing PWFs and EWFs under different conditions (Line 505-521). "Therefore, the utilisation efficiency of green water resources should be improved through water retention by tillage system and mulching. Meanwhile, more blue water can be generated through rainwater harvesting (Hoekstra, 2019). Specifically, we suggest two measures to increase the blue water efficiency in northern China. One is the rainwater harvesting in rainy season, especially for the short-time heavy rain which cannot effectively used by crops but easily cause soil erosion. The other one is reducing blue water consumption and loss at field by popularizing water-saving irrigation techniques and mulching practices. Such measure is helpful to improve the utilisation efficiency of both blue and green water. Based on the current results, we recommend the government to improve agricultural water use efficiency through the extension of water-saving irrigation techniques and better agricultural inputs management, especially in northwest China. High water consumption and low economic value crops' acreages in non-primary production areas should be reduced. For the southern regions with abundant rainwater resources, the economic benefits of irrigation are very limited, on the contrary, rainfed agriculture has obvious advantages and the potential to increase economic benefits. Therefore, farmers should improve the water conservation rate and the utilization efficiency of green water through farming system and coverage to reduce the amount of water used for irrigation. The government should also give financial subsidies for agricultural production to those provinces

where there were lose-lose relationships between reducing the water resources input for harvesting crop yields and optimizing the economic benefits per unit of water consumption. Finally, improve the field managements especially in utilization rate of chemical fertilizers and pesticides to increase agricultural productivity further (Zhang et al., 2013)."

7. In table 8, you are comparing EWF in 'Wheat in Morocco', 'Wheat in Tunisia', 'Winter wheat in China' and 'Spring wheat in China'. Considering the large differences in the regions/countries, is this possible to compare? Please also make note these comparisons are not in the same regions and make sure you include the assumptions you make in these comparisons.

**Response:** Yes, we realize that the current writing is not clear in related to the current Table 6. To be clearer, we added (Line 486-488) "There were no existing EWF values for China's cases. We wish to show the available values on EWF of crops, while for countries other than China."

8. The main goal of this paper is the SI, but I am still unclear on this index. I think this index needs to be explained in greater detail in the methodology as well as the interpretation and impact of this index in the subsequent results and discussion sections needs further improvement. This is the innovation of your paper so this needs to be more clear.

**Response:** As we respond to Referee #1's similar comments, we rewrote the following sentences:

In the start Section 2.5 we added the definition of the SI (Line 211-214). "The synergy evaluation index (SI) in the current study is the measure of the synergy levels between the PWF and EWF of crops, by summing up their corresponding difference between the water footprint and the base value divided by the range (the maximum minus the minimum) of the water footprint. Here, we adopt the national average level water footprint value as the reference for comparison."

We also highlight the meaning of SI again in the end Discussion (Line 542-547) as "In addition, it should be noted that in the current study, the SI measures, considering the spatial heterogeneities in crop WFs among provinces, the synergy levels between the current PWF and EWF. The synergy (both the PWF and EWF are lower than the national averages), trade-off (one is higher than the national average while the other is lower), or lose-lose (both are higher than the national averages) situation can be identified. The most optimized situation means high economic value generated by low water consumption. For the two provinces with high SI values, they were both in an advantageous position, while the one with a higher SI values performed better in terms of synergy between PWF and EWF."

9. You need to be careful with some English terminology such as 'contradiction'. I do not believe that is what you meant in several places where you use it.

**Response:** We are very sorry for the incorrect wording. As we respond to the similar comment by Referee #1, we checked carefully through the text and corrected the word "contradiction" into "trade-off" or "lose-lose" accordingly.

Specific comments

P3L79 – Change 'which are respectively calculated from the daily green ($ET_{g[t]}$, mm) and blue evapotranspiration ($ET_{b[t]}$, mm)' to which are respectively calculated from the blue evapotranspiration ($ET_{b[t]}$, mm) and daily green evapotranspiration ($ET_{g[t]}$, mm)' as you use respectively and then change the order or green and blue.

**Response:** We corrected the sentence in the revision (Line 132).

P5L131 – you should define what economic benefit unit is.

**Response:** The economic benefit unit refers to crop price in the current study. We clarified the terminology in the revision (Line 147-148).

P7L184 – move the data section before your methods.

**Response:** We showed the data sources by calculation steps. For example, the "AquaCrop model" was directly shown and readers will feel hard to follow without information on introduction of the model. Based on carefully consideration, we would keep the section Data following the method. We appreciate very much for your kind understanding.

P7L175 – You state 'Obviously, -2 <= SI <= 2'. Why is this obvious? Please explain and clarify in text.

**Response:** The synergy evaluation index (SI) is the sum of the difference between the water footprint and the base value divided by the range (the maximum minus the minimum) of the water footprint. Here, we adopt the national average level water footprint value as the reference of comparison. The SI is calculated as follows:

[revised manuscript text omitted]

---

## Author Response (AR2)

**Physical versus economic water footprints in crop production: a spatial and temporal analysis for China**

Xi Yang, La Zhuo, Pengxuan Xie, Hongrong Huang, Bianbian Feng, Pute Wu

**Authors' responses to Editor's comments**

We thank Editor prof. Ann van Griensven very much for the positive comments and valuable suggestions. Please find below our detailed responses point by point. The revised parts are coloured in RED in the revised manuscript.

Comments to the Author:

I have few comments.

1) I would suggest to replace 'study' by spatial and temporal analysis' in the title.

**Response:** The title is updated as "Physical versus economic water footprints in crop production: a spatial and temporal analysis for China".

2) Abstract In the abstract, line 17, explain the meaning of the synergy evaluation index (to reveal spatial autocorrelations?). I would also do it in line 77.

**Response:** As we mentioned in the text that the synergy evaluation index is constructed to reveal the synergies and trade-offs of crop water productivity and its economic value from the WF perspective. To be clearer, we revise the sentences in Abstract (Line 17-18) and Introduction (Line 77-78).

3) Line 79 you should not refer to the table, but this should be done in the methodology section.

**Response:** The pointed sentence in the Introduction is deleted. Instead, in Section 2.2 we add the sentences (Line 128-129) as "The PWF of 14 major crops were calculated annually in 31 provinces at the meteorological station level. Table 1 shows the number of meteorological stations per province."

4) line 85 remove --, and replace by 'which is a'.

**Response:** Yes, the correction is made.

*We have checked carefully in the revision for typos, missing co-authors and their affiliations, terminology, updates of data in tables, or updates of variables in equations. The missing corresponding author information has been added in the revision. We appreciate again for Editor's efforts on improving the study.

[revised manuscript text omitted]

$\qquad Y = f_{HI} HI_0 B$ , $\qquad\qquad\qquad\qquad\qquad\qquad\qquad\qquad\qquad\qquad\qquad\qquad\qquad$ (8)

where the adjustment factor($f_{HI}$) reflects the water and temperature stress depending on the timing and extent during the crop
cycle.

**2.2 Calculation of production-based water footprint (PWF)**

The PWF of 14 major crops were calculated annually in 31 provinces at the meteorological station level. Table 1 shows the
number of meteorological stations per province. The PWF (m$^3$ kg$^{-1}$) consists of the blue PWF (PWF$_b$, m$^3$ kg$^{-1}$) and the green
PWF (PWF$_g$, m$^3$ kg$^{-1}$), which are respectively calculated from the daily blue evapotranspiration ($ET_{b[t]}$, mm) and daily green
evapotranspiration ($ET_{g[t]}$, mm) and crop yield (Y, kg ha$^{-1}$) during the growing period (Hoekstra et al., 2011), as shown in Eqs.
(9) - (11):

$\qquad PWF = PWF_b + PWF_g$ , $\qquad\qquad\qquad\qquad\qquad\qquad\qquad\qquad\qquad\qquad\qquad$ (9)

[revised manuscript text omitted]